# PRL2 regulates neutrophil extracellular trap formation which contributes to severe malaria and acute lung injury

Xinyue Du [1,2,6], Baiyang Ren [1,3,6], Chang Li[1,3], Qi Li [1], Shuo Kan [1], Xin Wang [1,3], Wenjuan Bai [1], Chenyun Wu [1], Kokouvi Kassegne [2], Huibo Yan [1], Xiaoyin Niu[1], Min Yan [3], Wenyue Xu [4], Samuel C. Wassmer [5], Jing Wang [1] ✉, Guangjie Chen [1] ✉ & Zhaojun Wang [1,2] ✉

Excessive host immune responses contribute to severe malaria with high mortality. Here, we show that PRL2 in innate immune cells is highly related to experimental malaria disease progression, especially the development of murine severe malaria. In the absence of PRL2 in myeloid cells, *Plasmodium berghei* infection results in augmented lung injury, leading to significantly increased mortality. Intravital imaging revealed greater neutrophilic inflammation and NET formation in the lungs of PRL2 myeloid conditional knockout mice. Depletion of neutrophils prior to the onset of severe disease protected mice from NETs associated lung injury, and eliminated the difference between WT and PRL2 CKO mice. PRL2 regulates neutrophil activation and NET accumulation via the Rac-ROS pathway, thus contributing to NETs associated ALI. Hydroxychloroquine, an inhibitor of PRL2 degradation alleviates NETs associated tissue damage in vivo. Our findings suggest that PRL2 serves as an indicator of progression to severe malaria and ALI. In addition, our study indicated the importance of PRL2 in NET formation and tissue injury. It might open a promising path for adjunctive treatment of NET-associated disease.

Malaria is one of the most important tropical diseases, leading to great burdens on global health. In 2022, there were an estimated 249 million new cases of malaria and 608 000 malaria-related deaths in 85 countries[1]. Severe malaria is mainly caused by *Plasmodium falciparum* in which cerebral malaria, severe anemia, and acute respiratory distress syndrome/acute lung injury (ARDS/ALI) are common complications[2,3]. In the development of severe malaria syndromes, infected red blood cell (iRBC) sequestration in microvascular beds is a critical factor[4]. Meanwhile, excessive host immune responses also contribute to the pathogenesis of severe malaria[5].

Neutrophils are the most abundant innate immune effecter cells of the immune system that defend the host and clear pathogens by phagocytosis, degranulation and neutrophil extracellular trap (NET) release[6,7]. NETs are extracellular strands of decondensed DNA in complex with histones and neutrophil granule proteins, such as myeloperoxidase (MPO) and neutrophil elastase (NE)[7,8]. Stimulation of neutrophils with pathogenic or non-pathogenic components can lead to the activation of nicotinamide adenine dinucleotide phosphate (NADPH) oxidase via the protein kinase C signaling pathway[9]. Then, reactive oxygen species (ROS) are produced to activate protein

[1]Shanghai Institute of Immunology, Department of Immunology and Microbiology, Shanghai Jiao Tong University School of Medicine, Shanghai 200025, P.R. China. [2]Key Laboratory of Parasite and Vector Biology, Ministry of Health, China; School of Global Health, Chinese Center for Tropical Diseases Research, Shanghai Jiao Tong University School of Medicine, Shanghai 200025, P.R. China. [3]Department of Pathogen Biology and Immunology, Faculty of Basic Medical Science, Kunming Medical University, Kunming 650500, P.R. China. [4]Department of Pathogenic Biology, Army Medical University (The Third Military Medical University), Chongqing 400038, P.R. China. [5]Department of Infection Biology, London School of Hygiene & Tropical Medicine, London WC1E 7HT, UK. [6]These authors contributed equally: Xinyue Du, Baiyang Ren. ✉e-mail: jingwang@shsmu.edu.cn; guangjie_chen@163.com; zjwang@sjtu.edu.cn

arginine deiminase type-4, resulting in chromatin decondensation and NET formation[10]. It was reported that neutrophils play an essential role in anti-malaria immune responses by killing the pathogen. However, neutrophils might also contribute to pathogenesis in malaria. Neutrophil activation was reported to be strongly associated with increased disease severity in human malaria[11–13]. Numbers of circulating neutrophils often rise in patients infected by *P. falciparum*, in contrast, those of lymphocytes usually decrease[14]. Levels of neutrophil chemokine IL-8, NE, proteinase-3 and circulating NETs were increased in patients with malaria. It was reported that NETs induced by heme and tumor necrosis factor-α (TNF-α) have a key role in malaria immunopathology and severe malaria[12,15]. Therefore, the down-regulation of NETs could serve as a novel approach to prevent severe malaria.

Phosphatase of regenerating liver 2 (PRL2), encoded by the *protein tyrosine phosphatase IVA 2* (*PTP4A2*) gene, belongs to PRLs family and PTPs subfamily[16]. Our previous study revealed the role of PRL2 in antibacterial immune response[17]. PRL2 is highly expressed in resting immune cells, but can be markedly degraded under inflammation by sensing oxidative stress[18]. The deficiency of PRL2 leads to increased production of ROS and enhanced bactericidal activity by regulating the GTPase Rac pathway[17]. To further demonstrate the role of PRL2 in innate immunity, we analyzed the transcription profile of peripheral myeloid cells from PRL2 knockout (KO) and wildtype (WT) mice. Kyoto encyclopedia of genes and genomes (KEGG) pathway analysis showed that genes with significantly altered expression levels were enriched in malaria related signaling pathways. In the present study, we explored the role of PRL2 in malaria by using a *P. berghei* ANKA (PbA) infected C57BL/6 J mouse model. Our results indicated that PRL2 in innate immune cells is highly related to experimental malaria disease progression, especially the development of the lung injury in murine severe malaria. This work reveals that PRL2 regulates neutrophil activation and NET accumulation via the Rac-ROS pathway. PRL2 might be targeted to prevent pathologic NET formation, thus applied in severe malaria or ALI associated therapy.

## Results

### Reduced PRL2 protein in myeloid cells is correlated with murine severe malaria

To establish a mouse model of malaria, C57BL/6 J mice were intraperitoneally inoculated with $1 \times 10^6$ PbA infected RBCs. During infection, parasitemia and anemia status were monitored through mouse tail blood analysis for fourteen consecutive days. Signs of disease development were scored, and mouse survival rates were recorded. Consistent with previous reports, severe malaria was induced in PbA infected C57BL/6 J mice. Mice began to show severe malaria signs with brain, lung, liver and kidney injury on the 7th day post infection (dpi) and a portion of mice (4 of 6 mice) died during the second week of PbA infection (Supplementary Fig. 1a–h). Brain histopathological examination showed that PbA infection caused a slightly increased number of microthrombi, with intravascular accumulation of leukocytes and iRBCs at day 7 (Supplementary Fig. 1e, f). Meanwhile, histological sections of lung tissue revealed an intense inflammatory infiltrate, along with numerous neutrophils (Supplementary Fig. 1g, h). Flow cytometry analysis of peripheral blood cells from PbA infected mice (Day 7) showed a significantly greater increase in the CD11b+ myeloid population than that in uninfected control mice, as well as increased inflammatory cytokines levels (Supplementary Fig. 1i, j). To test whether PRL2 is associated with malaria, we measured the protein levels of PRL2 in different peripheral cell subsets in PbA infected and uninfected control mice by flow cytometry. As shown in Fig. 1a and Supplementary Fig. 1k, PRL2 protein levels were comparable in lymphocytes (B220+ and CD3+ cells) but decreased in myeloid cells (CD11b+ cells). We then investigated whether PRL2 is linked to malaria severity. We analyzed the correlation between the protein level of PRL2 and the clinical score of disease in PbA infected mice. Spearman's correlation analysis

indicated that the levels of PRL2 in CD11b+ myeloid cells were significantly decreased with increasing clinical scores (Fig. 1b, Supplementary Fig. 1l). Further analysis showed that mice with severe malaria had significantly decreased protein levels of PRL2 compared to mice with uncomplicated disease (Fig. 1c, Supplementary Fig. 1m). The above results indicated that reduced PRL2 protein in myeloid cells is associated with murine severe malaria.

### PRL2 deficiency in myeloid cells promotes the development of experimental severe malaria with lung injury

To explore how reduced PRL2 protein in myeloid cells is involved in the pathogenesis of malaria, we induced disease in PRL2 myeloid cell conditional knockout (CKO) mice and their WT littermates. PbA infection caused more severe disease in PRL2 CKO mice than in WT control mice. As shown in Fig. 1d, higher clinical scores and greater mortality were observed in PRL2 CKO mice. All the CKO mice died within nine days, whereas 50% of the WT mice remained alive until fourteen days after PbA infection. Severe anemia, cerebral malaria, acute lung injury and kidney injury are common severe malaria complications[4,5]. We analyzed the number of iRBCs and RBCs in PRL2 CKO and WT mice. There was no significant difference between the two groups (Fig. 1e), suggesting enhanced disease severity in CKO mice was not associated with parasitemia or anemia. We also compared brain pathology, adherence of iRBCs and leukocytes to brain vessels and vascular plugging in WT and CKO mice were observed on the 7th dpi. Although there was slightly increased in CKO mice, the difference between two groups was not significant (Supplementary Fig. 2a, b). The degree of blood-brain barrier (BBB) permeability, and the degree of liver or kidney impairment were also similar in the two groups (Supplementary Fig. 2c–e). However, the degree of lung injury showed remarkable differences between two groups. Images from computed tomography (CT) showed a remarkable increase in lung density of CKO mice on the 7th dpi, which indicating greater lung injury (Fig. 1f, g). Quantification of the pathological scores indicated that PRL2 CKO mice had thicker alveolar septa, aggravated intra-alveolar erythrocyte extravasation and alveolar infiltration of leukocytes, including greater neutrophil accumulation in the lungs (Fig. 1h, i). PRL2 CKO mice are more susceptible to severe malaria-associated ALI.

Excessive host immune responses play important roles in the pathogenesis of severe malaria, especially in malaria-associated ALI[19–21]. Therefore, we measured the immune cell population and cytokines in PbA infected CKO and WT mice. Deficiency of PRL2 in myeloid cells had no influence on the proportion and count of peripheral immune cells (Fig. 1j). Similarly, there was no difference in cytokine or chemokine responses (CCL-2, GM-CSF, IFN-β, IFN-γ, IL-1α, IL-6, IL-10 and TNF-α) between the two groups on the 7th dpi (Fig. 1k). However, Giemsa staining of blood smears and immunofluorescence showed that the number of NETs was significantly increased in PRL2 CKO mice compared with WT controls (Fig. 1l, m, Supplementary Fig. 2f). Serum NET levels in PRL2 CKO mice were dramatically higher than those in WT mice, which was evaluated by detecting the circulating MPO-DNA complex (Fig. 1n).

### PRL2 deficiency exacerbates malaria-induced ALI by promoting neutrophil infiltration and NET accumulation in lung tissue

NETs in the lungs on the 7th dpi were identified by immunofluorescence staining for MPO and citrullinated histone H3 (Cit-H3), which can specifically mark NETs. Statistical results showed that the numbers of neutrophils and NETs both increased in the lungs of PRL2 CKO mice (Fig. 2a, b). Moreover, we performed multiphoton intravital imaging to track neutrophils in the lungs of PbA infected mice. PE conjugated Ly6G antibody and Sytox Green (SG) were used to probe neutrophils and visualize extracellular DNA. Significantly more circulating neutrophils appeared in the lungs of CKO mice (Fig. 2c, Supplementary Fig. 3, Supplementary Movie 1, 2). NETs were observed in the

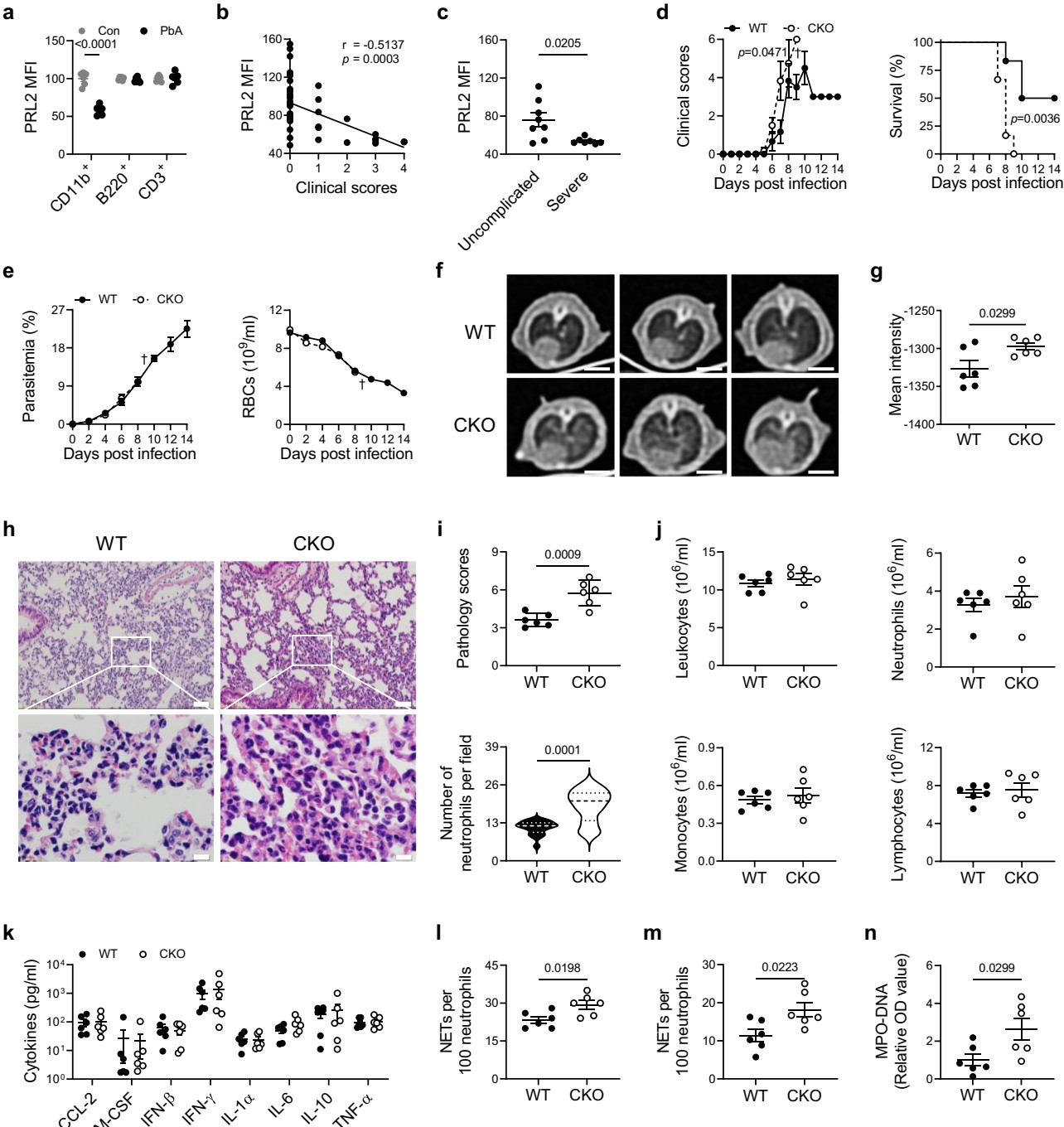

**Fig. 1 | PRL2 downregulation in myeloid cells is associated with severe malaria.**
**a**–**c** C57BL/6 J mice were infected with $1 \times 10^6$ *P. berghei* ANKA (PbA) iRBCs.
**a** Relative PRL2 mean fluorescence intensity (MFI) in different subsets of peripheral blood cells from the control (Con) and PbA infected mice at 7 days post infection (dpi), normalized to normal mice ($n = 6$ mice per group). **b** Spearman's correlation analysis between PRL2 MFI in CD11b⁺ peripheral blood cells and the clinical scores of PbA infected mice ($n = 46$, samples from 10 mice at 0, 2, 4, 6, 8 dpi). **c** Relative PRL2 MFI in CD11b⁺ peripheral blood cells from mice with uncomplicated and severe malaria, samples as in (**b**). **d**, **e** Wildtype (WT) and PRL2 myeloid cell con-ditional knockout (CKO) mice were infected with $1 \times 10^6$ PbA iRBCs. **d** Clinical scores, survival curve, (**e**) parasitemia and number of RBCs during the 14-day infection ($n = 6$ mice per group). Cross symbol indicates all CKO mice died. **f**–**n** WT and PRL2 CKO mice were infected with $1 \times 10^6$ PbA iRBCs ($n = 6$ mice per group). Samples were collected at 7 dpi. **f** Representative axial computed tomography (CT) images of lungs. Scale bars, 5 mm. **g** Quantification of mean pulmonary density

from (**f**). **h** Hematoxylin-eosin (H&E) staining of lung sections. Representative images are shown. Scale bars, up: 50 μm, down: 10 μm. **i** Pulmonary pathology scores and infiltrated neutrophils numbers were quantified from (**h**). **j** Numbers of peripheral leukocytes, neutrophils, monocytes and lymphocytes. **k** Concentrations of serum cytokines and chemokines. **l** Quantification of peripheral neutrophil extracellular traps (NETs) via an analysis of Giemsa-stained blood smears. **m** Quantification of peripheral NETs via immunofluorescence co-staining with myeloperoxidase (MPO) and citrullinated histone H3 (Cit-H3). **n** Relative OD value of serum MPO-DNA level. All data were pooled from two independent experiments. Data are presented as the mean ± SEM or in violin plots showing the median and interquartile range. *p* values were calculated by two-tailed unpaired *t* test (**a**, **d** left, **g**, **i** up, **l**, **m**, **n**), two-tailed Mann–Whitney test (**c**, **i** down), Spearman's correlation (**b**) or log-rank test (**d** right) and shown in the figures. Source data are provided as a Source Data file.

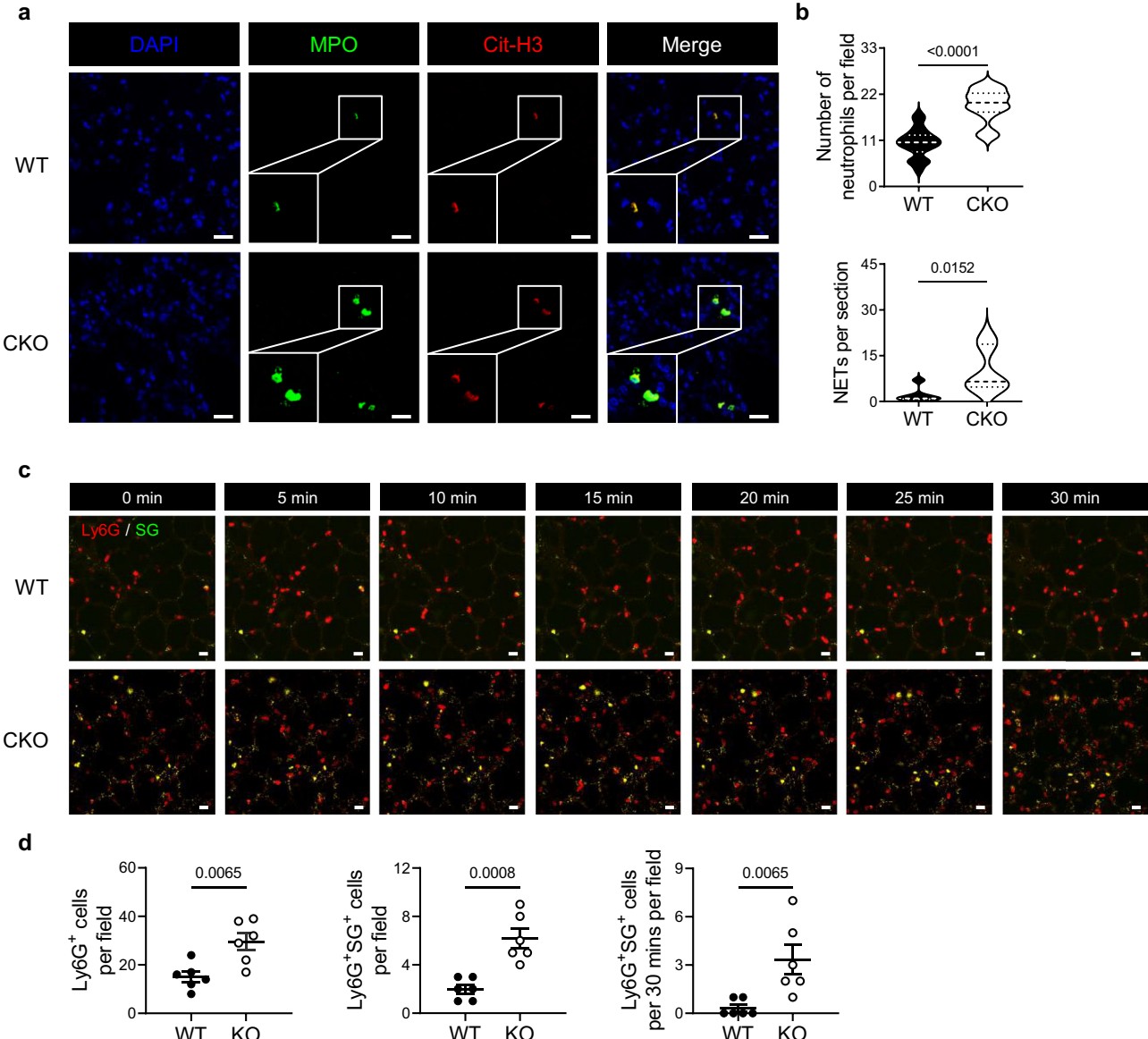

**Fig. 2 | Mice with PRL2 myeloid cell deficiency develop more severe malaria related lung injury with increased NET accumulation.** Wildtype ( WT) and PRL2 myeloid cell conditional knockout (CKO) mice were infected with $1 \times 10^6$ *P. berghei* ANKA (PbA) iRBCs. **a** Representative immunofluorescence images of lungs from WT and PRL2 CKO mice at 7 dpi ($n = 6$ mice per group). DNA is stained in blue (DAPI), myeloperoxidase is stained in green (MPO) and citrullinated histone H3 is stained in red (Cit-H3). Scale bars, 20 μm. Neutrophils are indicated as co-stained with MPO and DAPI. Neutrophil extracellular traps (NETs) are indicated as co-stained with MPO and Cit-H3. **b** Quantification of neutrophils and NETs from (**a**). **c** Intravital imaging of the lungs from WT and PRL2 CKO mice at 7 dpi ($n = 6$ mice per group). Representative time-lapse images are shown. Scale bars, 20 μm. Neutrophils are stained in red (Ly6G) and extracellular DNA is stained in green (Sytox Green, SG). **d** Quantitative analysis of Ly6G+ cells, Ly6G+SG+ cells, and Ly6G+SG+ cells/30 min in each field of each mice lung by intravital imaging. All data were pooled from two independent experiments. Data are presented as the mean ± SEM or in violin plots showing the median and interquartile range. *p* values were calculated by two-tailed unpaired *t* test (**b** up, **d** left and middle) or two-tailed Mann–Whitney test (**b** down, **d** right) and shown in the figures. Source data are provided as a Source Data file.

pulmonary capillaries of both WT and CKO mice with PbA. The number of NETs per field of view was significantly greater in CKO mice than in control mice (Fig. 2d). These data demonstrated that the deficiency of PRL2 in myeloid cells exacerbates malaria-induced ALI by promoting the accumulation of NETs in lung tissues.

To further investigate the role of PRL2, neutrophils, and NETs in the development of ALI in severe malaria, PbA infected WT and PRL2 myeloid CKO mice received a single dose of anti-Ly6G monoclonal antibody or control antibody prior to the onset of severe disease (Fig. 3a). The depletion of neutrophils was effective on the second day after antibody treatment (Fig. 3b). Single dose anti-Ly6G treatment on the 6th dpi didn't show significantly effects on parasitemia and anemia

degree in PbA infected WT and CKO mice (Supplementary Fig. 4). However, neutrophil depletion at this late stage of the infection successfully increased survival rate of PRL2 CKO mice and abolished the difference between WT and CKO mice (Fig. 3c). Remarkably, anti-Ly6G treatment reduced lung pathology associated with neutrophil infiltration and NET accumulation, thereby abrogating the difference observed between WT and PRL2 CKO group. (Fig. 3d–g).

**PRL2 controls NET formation via regulating Rac GTPase activation and ROS production**

To directly test the effect of PRL2 on NET formation, we purified primary WT and PRL2 KO neutrophils, and then challenged them

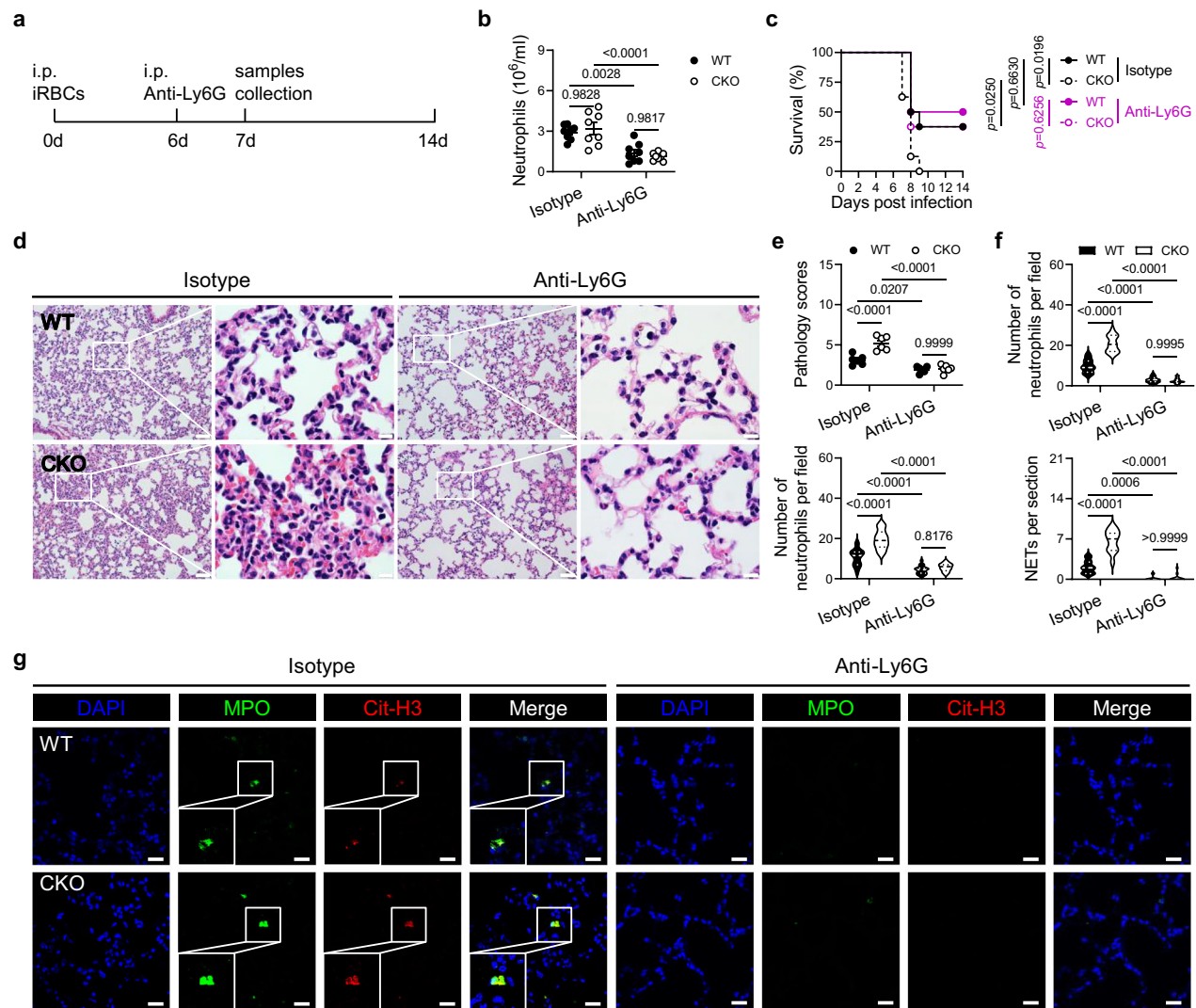

**Fig. 3 | Depletion of neutrophils protects PRL2 myeloid cell deficient mice from severe malaria related acute lung injury. a** A schematic showing the experimental design for neutrophil depletion in *P. berghei* ANKA (PbA) infection model. PbA infected wildtype (WT) and PRL2 myeloid cell conditional knockout (CKO) mice were intraperitoneally injected with anti-Ly6G monoclonal antibody or an isotype control at 6 days post infection (dpi). **b** Number of neutrophils in peripheral blood from the four groups of mice described in (**a**) at 7 dpi (*n* = 8 mice per group). **c** Survival curve of the mice described in (**b**). **d** Hematoxylin-eosin (H&E) staining of histologic lung sections from the four groups of mice described in (**a**) at 7 dpi (*n* = 6 mice per group). Representative images are shown. Scale bars, left: 50 μm, right: 10 μm. **e** Pulmonary pathology scores and infiltrated neutrophils numbers were

quantified from (**d**). **f** Quantification of neutrophils and neutrophil extracellular traps (NETs) in immunofluorescence images of lungs from the four groups of mice described in (**d**) at 7 dpi. **g** Representative immunofluorescence images of lungs as described in (**f**). DNA is stained in blue (DAPI), myeloperoxidase is stained in green (MPO) and citrullinated histone H3 is stained in red (Cit-H3). Scale bars, 20 μm. Neutrophils are indicated as co-stained with MPO and DAPI. NETs are indicated as co-stained with MPO and Cit-H3. All data were pooled from two independent experiments. Data are presented as the mean ± SEM or in violin plots showing the median and interquartile range. *p* values were calculated by two-way ANOVA with Tukey's multiple testing (**b, e, f**) or log-rank test (**c**) and shown in the figures. Source data are provided as a Source Data file.

with phorbol myristate acetate (PMA), a well-known NET inducer. PMA stimulation induces neutrophils to release spiky NETs. Notably, neutrophils with PRL2 deficiency showed more pronounced morphological changes and a higher ratio of NETs than WT cells (Fig. 4a, b, Supplementary Fig. 5a, b). In malaria, NETs are induced by heme, a well-known damage associated molecular pattern (DAMP), released by iRBCs[22]. Heme is easily oxidized to form hemin, which is a type of porphyrin containing chlorine and is used in the laboratory[23]. Thus, we exposed WT and PRL2 KO neutrophils to iRBC and hemin. Both iRBC and hemin induced NET formation, but PRL2 deficient cells formed significantly more NETs after iRBC or hemin stimulation (Fig. 4a, b, Supplementary Fig. 5a, b). We also performed live-cell imaging. Equivalent amount of WT and PRL2 deficient TdTomato[+] neutrophils were mixed and then stimulated with

PMA together. Consistent with the above results, PRL2 KO neutrophils released much more NETs than WT cells (Fig. 4c, d, Supplementary Movie 3).

ROS are critical for NET formation[7]. Therefore, we used a luminol dependent chemiluminescence assay to measure ROS production. In response to PMA stimulation, neutrophils quickly increased ROS generation. The peak values of chemiluminescence were observed within one hour of stimulation. PRL2 deficiency increased ROS production (Fig. 5a). Similar results were found when intracellular ROS were measured by CM-H$_2$DCFDA staining (Fig. 5b). Furthermore, the antioxidant N-acetylcysteine (NAC) inhibited NET release and abrogated the difference between WT and PRL2 KO cells (Fig. 5c). The above results indicated that the effect of PRL2 on NET formation was dependent on ROS.

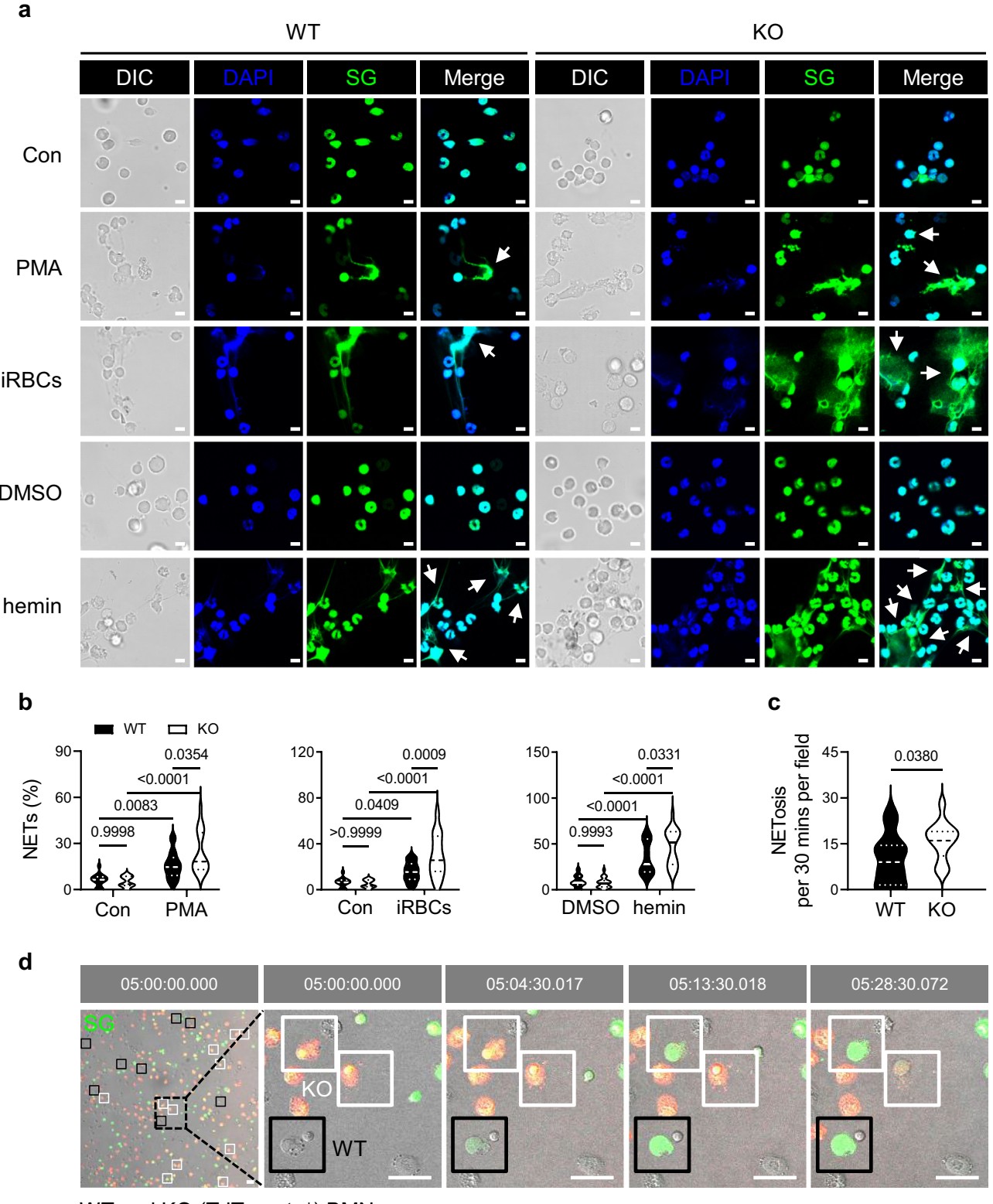

**d** WT and KO (TdTomato⁺) BMN

In neutrophils, ROS are mainly generated by the NADPH oxidase complex, which is made up of five oxidase units and a Rac GTPase. PRL2 binds to and regulates the activity of Rac GTPase[17]. Therefore, we asked whether PRL2 regulates ROS dependent NET formation through Rac GTPase. To address this question, we first used a PAK pulldown assay to test the effect of PRL2 deficiency on Rac activation in neutrophils. We found that activated Rac GTPase was much more strongly induced in PRL2 KO neutrophils (Fig. 5d). Second, we detected the activation of Rac GTPase by incubating the cells with Rac-GTP antibody and imaged by confocal microscopy. Consistently, immunofluorescence data showed stronger activation of Rac in PRL2 KO cells (Fig. 5e, f). In addition, pretreatment with the Rac inhibitor NSC23766 prevented NET formation induced by PMA, iRBCs or hemin, and blocked the differences between cells with and without PRL2 (Fig. 5g).

**Fig. 4 | PRL2 deficiency promotes NET formation. a** Representative immuno-fluorescence images of neutrophil extracellular traps (NETs) released by wildtype (WT) and PRL2 knockout (KO) bone marrow derived neutrophils (BMNs) after phorbol myristate acetate (PMA), iRBC and hemin stimulation. Fixed cells were stained. DNA is stained in blue (DAPI), and extracellular DNA is stained in green (Sytox Green, SG). Scale bar, 5 μm. DIC, differential interference contrast. Arrowheads indicate NETs. **b** Quantification of NETs in (**a**). Data were pooled from three independent experiments. BMNs were collected from different mice (*n* = 3 per group). **c** Equivalent amount of WT (in white) and PRL2 deficient TdTomato⁺ (in red) neutrophils were mixed and then stimulated with PMA together for 5 h.

Extracellular DNA is stained in green (SG). Quantitative analysis of NETosis/30 min in each field by live cell imaging. Data were pooled from three independent experiments. BMNs were collected from different mice (*n* = 3 per group). **d** Representative time-lapse images from live cell imaging described in (**c**) are shown. Black/white boxes indicate NETs from WT/KO BMNs, respectively. Scale bars, 25 μm. Violin plots show the median and interquartile range. *p* values were calculated by two-way ANOVA with Tukey's multiple testing (**b**) or two-tailed unpaired *t* test (**c**) and shown in the figures. Source data are provided as a Source Data file.

Together, these results suggest that PRL2 regulates NET formation via the Rac-ROS pathway.

## PRL2 negatively regulates NET formation in acute lung injury

To further determine the function of PRL2 in NET formation and its associated tissue injury, we established two ALI models. First, we mimicked the inflammatory environment of severe malaria, and set up a 24-h murine model of ALI using iRBCs combined with TNF-α (Fig. 6a). WT mice were injected with iRBCs 16 h prior to TNF-α treatment. The lungs and blood samples of these mice were harvested 8 h after TNF-α administration. Hematoxylin and eosin (H&E) staining clearly indicated that neutrophil infiltration and tissue damage were present within the pulmonary tissues (Fig. 6b, c). Accumulations of NETs were also observed by MPO and Cit-H3 staining (Fig. 6d, e). Accompanying with the lung injury, an increased percentage of CD11b⁺ myeloid cells and dramatically reduced PRL2 protein levels were observed in peripheral CD11b⁺ myeloid cells, especially in CD11b⁺Ly6G⁺ neutrophils (Fig. 6f, g). We then induced ALI in WT and PRL2 CKO mice. As shown in Fig. 6h−m, CKO mice showed significantly more severe lung injury and increased pulmonary NETs than WT mice. Depletion of neutrophils protect mice from ALI and reduce the difference between WT and PRL2 CKO mice. The above results indicated that reduced PRL2 is responsible for the NET accumulation and malaria-associated ALI.

Second, lipopolysaccharide (LPS) was used to induce ALI in mice (Supplementary Fig. 6). Consistent results were observed in this widely used ALI model, suggesting the important role of PRL2 in NET-associated ALI.

## Hydroxychloroquine ameliorates ALI via blocking PRL2 degradation and NET accumulation

Evidence indicates that hydroxychloroquine (HCQ) inhibits inflammation-induced PRL2 degradation[18], and this phenomenon can also be observed in neutrophils (Supplementary Fig. 7). We hypothesized that HCQ administration could prevent PRL2 reduction and protect mice from NET-associated ALI. We examined the potential relationship between HCQ and PRL2 in the iRBC and TNF-α induced ALI model first (Fig. 7a). There was no difference between ALI mice in peripheral RBCs and parasitemia when treated with normal saline (NS) or HCQ (Supplementary Fig. 8). HCQ had no significant effect on the percentages of myeloid cells in the blood of ALI mice (Fig. 7b). However, PRL2 protein levels in myeloid cells (CD11b⁺), especially in neutrophils (CD11b⁺Ly6G⁺), were reversed in ALI mice after HCQ treatment (Fig. 7c). H&E staining results showed that HCQ treatment significantly alleviated neutrophil infiltration and pathology of the tissues in ALI (Fig. 7d, e). In agreement with this decreased number of pulmonary neutrophils, the levels of NETs were significantly diminished after HCQ treatment (Fig. 7f, g). Similar results were obtained when ALI was induced by LPS (Supplementary Fig. 9). Thus, HCQ treatment blocked the PRL2 reduction in neutrophils and alleviated the pathology of the tissues in ALI.

## Discussion

Severe malaria is still a leading cause of morbidity and mortality in endemic areas[1,24]. When manifested in the lungs, severe malaria causes

ALI[20,25]. It has been reported that as many as 20–30% of severe malaria cases caused by *P. falciparum* or *P. viviax* lead to lung injuries[26]. Patients infected with *Plasmodium* can develop ALI with a mortality rate close to 80%[20]. The onset of ALI in malaria is sudden, progresses rapidly and can occur any time during *Plasmodium* infection. More importantly, it results in high mortality rates despite adequate therapeutic management[27,28]. In this study, we found that mice with severe malaria had significantly decreased PRL2 in blood myeloid cells. The levels of PRL2 were decreased with increasing clinical scores. These correlations suggested that PRL2 may serve as a diagnostic and prognostic biomarker for severe malaria.

Neutrophils are the most abundant leukocytes in the circulation. While they are known to play a pivotal role in the innate immune defense against malaria parasites, recent reports suggest they can also be involved in the development of complications during malaria infection[22]. Evidences indicate that neutrophil infiltration and NET formation lead to hyperinflammation in the lungs, and the rate of this infiltration and NET production are directly related to the severity of the disease[29]. In this study, we used PbA-infected C57BL/6 J mice. This well-established murine model of severe malaria causes multiple organ dysfunction[30,31]. Besides the brain, highly vascularized organs such as the lungs and kidneys are affected during PbA infection[32]. PbA infection caused more severe disease in PRL2 CKO mice than in WT control mice, and demonstrated that the loss of PRL2 predominantly leads to lung injury in this model. It could be due to the specific anatomical structure and microenvironment of lung tissues, as NETs can expand more easily in pulmonary alveoli, where they are able to directly induce epithelial and endothelial cell death. The detrimental effect of excessive NET release is therefore particularly pathogenic in the lungs[33]. NETs can be induced by heme, a well-known DAMP released by iRBCs in malaria[22]. Here, we demonstrate that PRL2 is involved in both iRBCs- and heme-induced NET formation. Further studies are warranted to investigate the nature of the parasite protein triggering this pathway.

Cellular ROS are essential for cellular signaling and cell survival. ROS may act as a rheostat allowing different cell death mechanisms to be engaged and crosstalk with different cell death types[34]. In neutrophils, ROS is required for NET formation with almost all stimuli[35]. ROS generated by NADPH oxidase stimulate myeloperoxidase (MPO), causing the activation and translocation of neutrophil elastase (NE) from neutrophil granules. In turn, the migration of elastase into neutrophil nuclei leads to chromatin decondensation[36]. Here, we show that PRL2 regulates NET formation via Rac-ROS pathway during experimental malaria infection. The accumulation of NETs contributes to the pathology of tissue damage in severe malaria and ALI in our murine model, and this phenomenon can be blocked by the depletion of neutrophils, even after the first signs of infection appear in animals. Besides NET formation, the high concentrations of ROS in neutrophils are also important for triggering apoptosis, necroptosis, necrosis, etc.[37]. Whether PRL2 regulates the development of severe malaria or ALI via neutrophils necrosis and necroptosis remains to be investigated. During inflammation, PRL2 can sense oxidative stress via highly reactive cysteine residues and is rapidly degraded through autophagy[18,38]. Malaria is a highly inflammatory and oxidative disease[39]. During the blood stage of malaria, various sources contribute to the

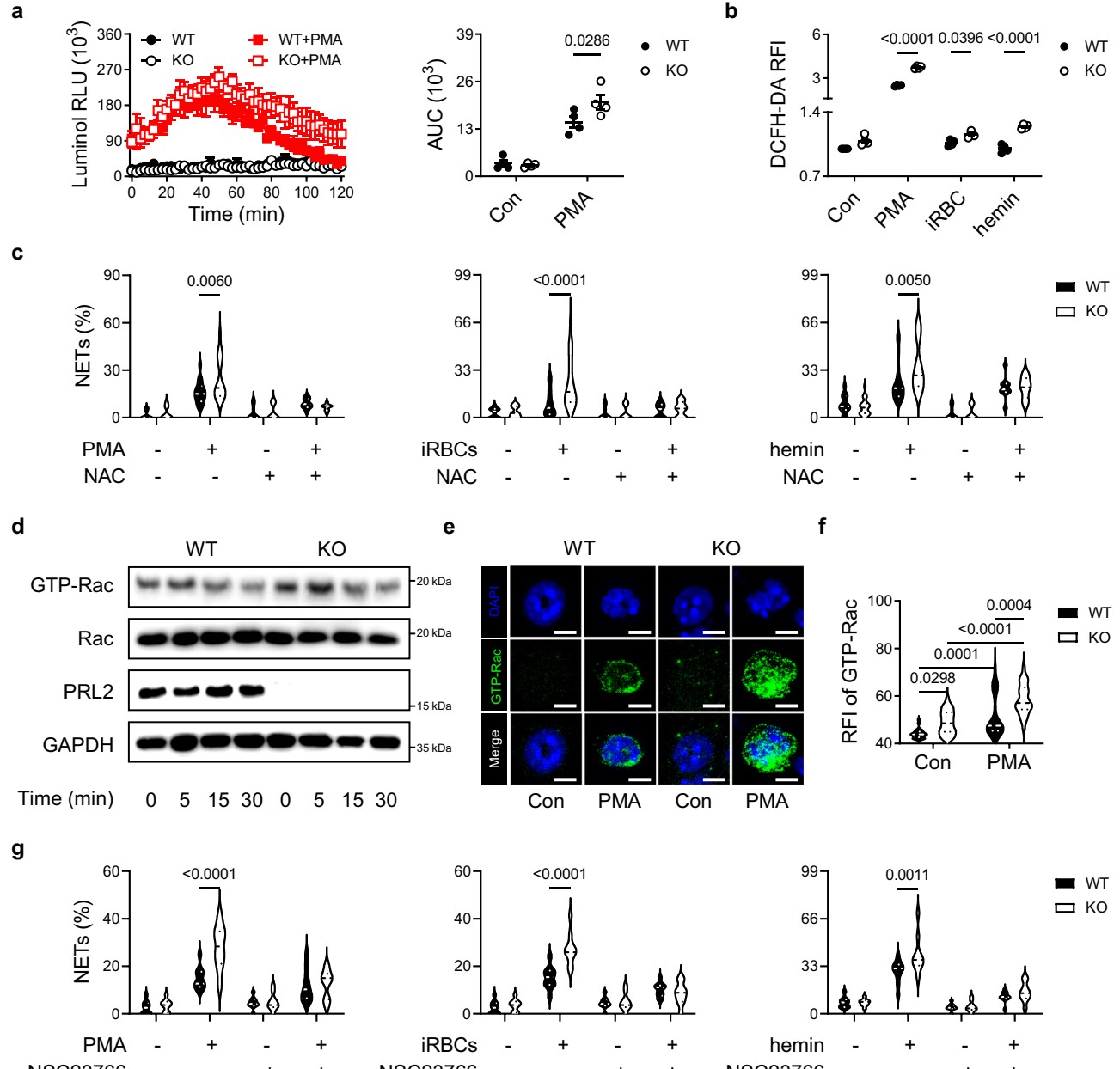

**Fig. 5 | PRL2 regulates NET formation through the Rac-ROS pathway. a** Wildtype (WT) and PRL2 knockout (KO) bone marrow derived neutrophils (BMNs) were stimulated with phorbol myristate acetate (PMA). Reactive oxygen species (ROS) production was measured by chemiluminescence assay. ROS kinetic plots and averaged area under the curve (AUC) are shown. **b** WT and PRL2 KO neutrophils were treated with PMA, iRBCs and hemin, ROS production was measured by fluorescent staining. **a, b** Data were pooled from four independent experiments. BMNs were isolated from different mice (*n* = 4 per group). **c** Quantification of neutrophil extracellular traps (NETs) released by WT and PRL2 KO neutrophils that stimulated with PMA, hemin and iRBCs. Cells were pretreated with NAC or not. Data were pooled from three independent experiments. BMNs were isolated from different mice (*n* = 3 per group). **d** WT and PRL2 KO neutrophils were treated with PMA for the indicated times. Cell lysates were subjected to pull-down using PAK GST beads. Lysates from pull-down and total lysates were subjected to SDS-PAGE followed by immunoblot assay with anti-Rac, anti-PRL2 and anti-GAPDH antibodies. The experiments were repeated two times with similar results, and the representative immunoblot images were shown. **e** Representative immunofluorescence images of WT and PRL2 KO neutrophils after PMA stimulation. The experiments were repeated three times with similar results. DNA is stained in blue (DAPI), and GTP-Rac is stained in green. Scale bar, 5 μm. **f** Relative fluorescence intensity (RFI) of GTP-Rac from WT or KO cells in (**e**) was determined (*n* = 26 cells per group). **g** Quantification of NETs released by WT and PRL2 KO neutrophils that stimulated with PMA, hemin and iRBCs. Cells were pretreated with NSC23766 or not. Data were pooled from three independent experiments. BMNs were from different mice (*n* = 3 per group). Data are presented as the mean ± SEM or in violin plots showing the median and interquartile range. *p* values were calculated by two-way ANOVA with Tukey's multiple testing (**a** right, **b**, **c**, **f**, **g**) and shown in the figures. Source data are provided as a Source Data file.

oxidative environment, including heme released from iRBCs or RBCs, the systemic upregulation of host oxidative enzymes and the oxidative burst from phagocytes[39]. In the PbA-infected murine malaria model, dramatically reduced PRL2 protein levels were observed in CD11b[+] myeloid cells. In ALI models, PRL2 protein was also decreased. HCQ is an FDA-approved antimalarial drug that acts as a lysosome inhibitor, and can also block inflammation-induced PRL2 degradation[18]. Several studies have proven that HCQ attenuates NET formation in patients with autoimmune disease and COVID-19[40]. In this study, we propose a mechanism whereby HCQ alleviates NET-associated pathology

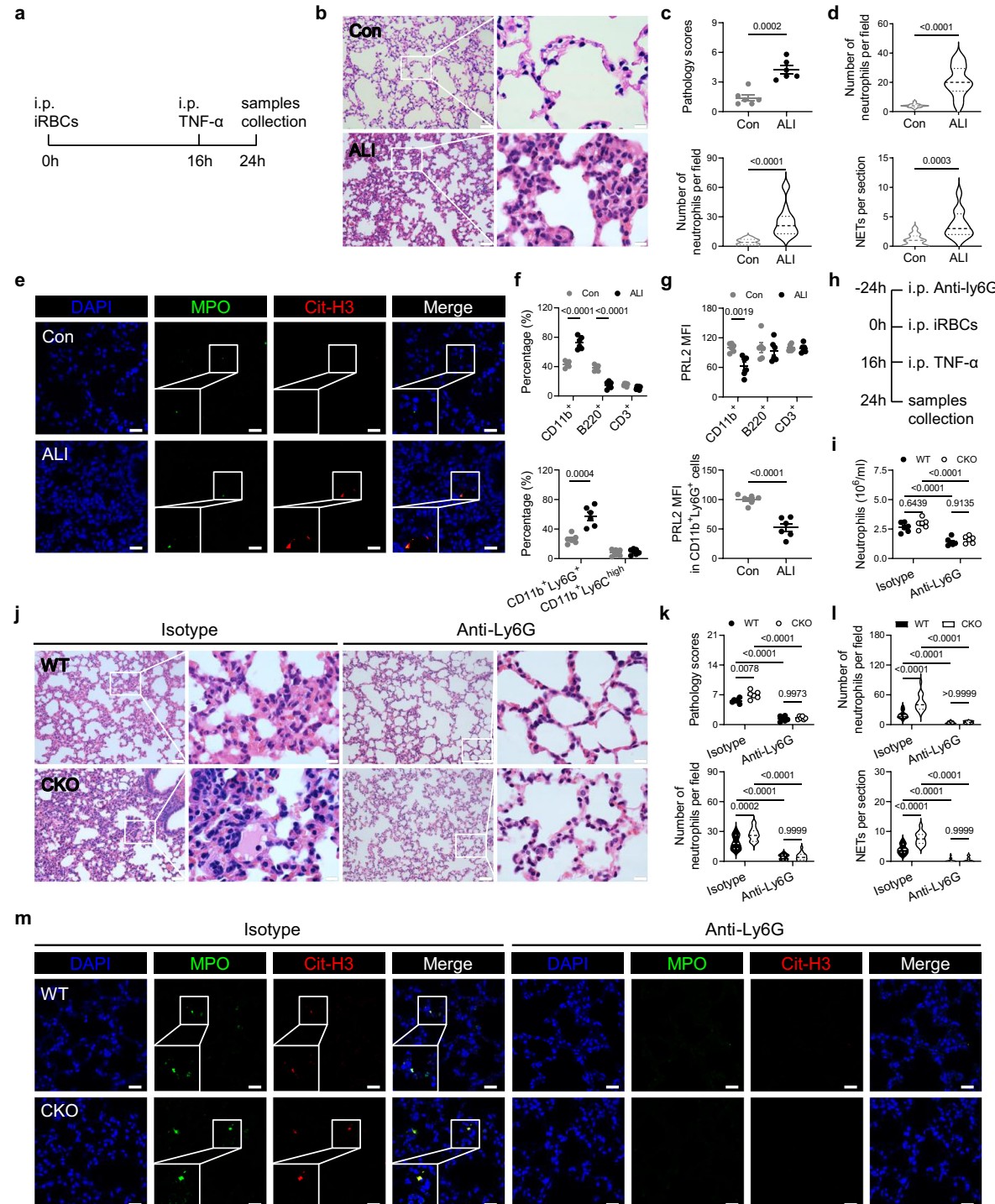

**Fig. 6 | PRL2 related NET formation contributes to the pathology in iRBCs induced ALI. a** A schematic showing the experimental design for iRBCs and TNF-α induced acute lung injury (ALI) model. **b**–**g** C57BL/6 J mice were induced ALI in (**a**) or grouped as normal control (Con) (*n* = 6 mice per group). **b** Representative images of lung tissues stained with hematoxylin-eosin (H&E). Scale bars, left: 50 µm, right: 10 µm. **c** Pulmonary pathology scores and infiltrated neutrophil numbers were quantified from (**b**). **d** Quantification of neutrophils and neutrophil extra-cellular traps (NETs) in immunofluorescence images of lungs. **e** Representative immunofluorescence images of lungs as described in (**d**). DNA is stained in blue (DAPI), myeloperoxidase is stained in green (MPO) and citrullinated histone H3 is stained in red (Cit-H3). Scale bars, 20 µm. Neutrophils are indicated as co-stained with MPO and DAPI. NETs are indicated as co-stained with MPO and Cit-H3. **f** Percentage of different subsets of peripheral blood cells. **g** Relative PRL2 mean fluorescence intensity (MFI) in different subsets of peripheral blood cells, nor-malized to normal mice. **h** A schematic showing the experimental design for

neutrophil depletion in iRBCs and TNF-α induced ALI model. **i**–**m** Wildtype (WT) and PRL2 myeloid cell conditional knockout (CKO) mice were intraperitoneally injected with anti-Ly6G monoclonal antibody or an isotype control 24 h before iRBCs injection (*n* = 6 mice per group). **i** Number of peripheral neutrophils from the four groups of mice 24 h after iRBCs injection. **j** Representative images of lung tissues stained with H&E. Scale bars, left: 50 µm, right: 10 µm. **k** Pulmonary pathology scores and infiltrated neutrophils numbers were quantified from (**j**). **l** Quantification of neutrophils and NETs in immunofluorescence images of lung tissues. **m** Representative immunofluorescence images of lung tissues. Staining panel is as described in (**e**). All data were pooled from two independent experi-ments. Data are presented as the mean ± SEM or in violin plots showing the median and interquartile range. *p* values were calculated by two-tailed unpaired *t* test (**c** up, **f**, **g**), two-tailed Mann–Whitney test (**c** down, **d**) or two-way ANOVA with Tukey's multiple testing (**i**, **k**, **l**) and shown in the figures. Source data are provided as a Source Data file.

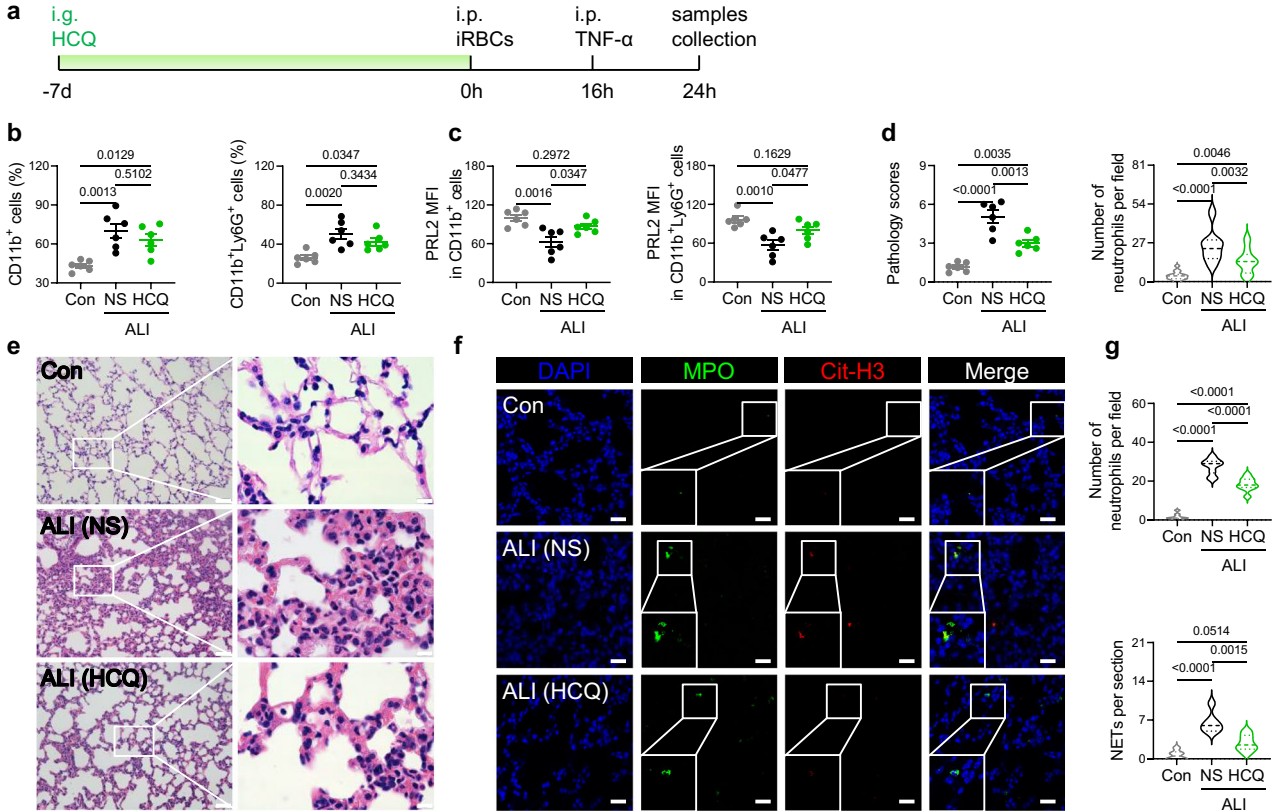

**Fig. 7 | HCQ alleviated iRBCs induced lung injury by blocking PRL2 degradation and NET formation. a** A flow chart depicting hydroxychloroquine (HCQ) treatment for iRBCs and TNF-α induced acute lung injury (ALI). **b**–**g** C57BL/6 J mice were treated with normal saline (NS) or HCQ and induced ALI, or just treated with NS and grouped as normal control (Con) (*n* = 6 mice per group). **b** Percentage of peripheral myeloid cells (CD11b[+]) and neutrophils (CD11b[+]Ly6G[+]).**c** Relative PRL2 mean fluorescence intensity (MFI) in different subsets of peripheral blood cells, normalized to normal control mice. **d** Pulmonary pathology scores and infiltrated neutrophils numbers were quantified from lung tissues stained with hematoxylin-eosin (H&E). **e** Representative images of lung tissues. Scale bars, left: 50 μm, right: 10 μm.

**f** Representative immunofluorescence images of lung tissues. DNA is stained in blue (DAPI), myeloperoxidase is stained in green (MPO) and citrullinated histone H3 is stained in red (Cit-H3). Scale bars, 20 μm. Neutrophils are indicated as co-stained with MPO and DAPI. Neutrophil extracellular traps (NETs) are indicated as co-stained with MPO and Cit-H3. **g** Quantification of neutrophils and NETs from (**f**). All data were pooled from two independent experiments. Data are presented as the mean ± SEM or in violin plots showing the median and interquartile range. *p* values were calculated by one-way ANOVA with Tukey's multiple testing (**b**, **c**, **d**, **g**) and shown in the figures. Source data are provided as a Source Data file.

through a blockade of PRL2 reduction. Our study indicates the importance of PRL2 in NET formation and tissue injury which upon validation in malaria patients, may open promising adjunctive therapy avenues for NET-associated pathology in malaria infection.

## Methods

### Ethics statement

Our research complies with all relevant ethical regulations. The animal study was reviewed and approved by the Animal Ethics Committee of Shanghai Jiao Tong University School of Medicine. All animal experiments were approved by the Institutional Animal Care and Use Committee (IACUC) of Shanghai Jiao Tong University School of Medicine (Project numbers A-2022-093 and A-2022-094).

### Mice

PRL2 myeloid cell conditional knockout (CKO, *Ptp4a2[fl/fl]LysM[Cre+]*) C57BL/6 J mice and their wild-type (WT, *Ptp4a2[fl/fl]*) littermates were generated as previously described[17]. Wild-type C57BL/6 J mice were purchased from Shanghai Lingchang Biotechnology Company Limited. B6-G/R mice (B6/JGpt-H11[em1Cin(CAG-LoxP-ZsGreen-Stop-LoxP-tdTomato)]/Gpt, Strain NO.T006163) were purchased from GemPharmatech Company Limited. PRL2 myeloid CKO-TdTomato[+] C57BL/6 J mice were generated by crossing PRL2 myeloid CKO mice and B6-G/R mice. Mice were housed in the Shanghai Jiao Tong University School of Medicine

Animal Care Facilities under specific pathogen-free conditions with 12-h light/dark cycle at 20–24 °C and 45–65% humidity.

### *Plasmodium berghei* ANKA infection

*P. berghei* ANKA (PbA) was provided by the Chinese Center for Disease Control and Prevention. Six- to eight-week old female mice were intraperitoneally injected with 1×10[6] PbA infected RBCs (iRBCs). The mortality of mice was monitored daily. Signs of disease development were scored daily as follows: 0 is no sign, 1 to ruffled fur, 2 to hunching, 3 to wobbly gait, 4 to limb paralysis, 5 to convulsions, 6 to coma. Mice with scores greater than or equal to 3 were considered to have severe malaria[41]. Blood samples were collected from the mouse tail every other day for determination of parasitemia and hematological parameters by Giemsa staining. To assess the permeability of blood–brain barrier (BBB), mice were injected intravenously with 2% Evans blue dye (BBI, A602025) 1 h prior to dissection at the indicated time. Serum alanine aminotransferase (ALT), aspartate aminotransferase (AST), uric acid (URIC) and blood urea nitrogen (BUN) were measured by Beckman-Coulter chemistry analyzer AU5800. Computed tomography (CT) of mice were performed by using IVIS Spectrum CT (PerkinElmer, USA). Lung density was quantified by ITK-SNAP software at 7 days post infection (dpi). An increase in lung density could indicate greater injury.

## Acute lung injury mouse models

For iRBCs induced acute lung injury (ALI) model, fresh blood was collected from donor mice which infected by PbA with parasitemia at 10%. Each six- to eight-week old male mouse received 500 µl blood immediately and 0.5 µg TNF-α 16 h later via intraperitoneal injection. Blood and lung were collected 24 h after the first injection.

For LPS induced ALI model, six- to eight-week old male mice were nasal administered with LPS solution (1.5 mg/kg) to induce ALI. Mice intranasally inoculated with an equal volume of PBS were used as a control. Blood and lung samples were collected 24 h later.

For HCQ treatment, a dose of 80 mg/kg/d HCQ (SPH Zhongxi Pharmaceutical Co., Ltd., China) was administered by oral gavage a week before acute lung injury model establishment. Mice received an equal volume of normal saline (NS) were used as a control.

## Neutrophil depletion

In vivo neutrophil depletion was performed by administering intraperitoneal (i.p.) injection of anti-Ly6G antibody (500 µg/dose, BioX-Cell, BE0075-1, clone 1A8) or an isotype control (500 µg/dose, BioXCell, BE0089, clone 2A3) at the indicated time. Blood samples were collected from the mouse tail for leukocyte quantification using blood smears.

## Histopathological analyses

For mice infected with PbA, brain and lung tissues were collected at 7 dpi. For mice with acute lung injury induced by iRBCs or LPS, lungs were collected 24 h after establishing the model. Tissues were fixed in 4% paraformaldehyde (PFA) for 48 h. After washing with phosphate-buffered saline (PBS), samples were embedded in paraffin, sectioned at 4 µm thicknesses, stained with hematoxylin-eosin (H&E) solution and examined by microscopy (Nikon Eclipse 80i) for histological changes. For brain tissues, leukocyte infiltration was identified in the blood vessels of brain sections to assess thrombotic events. For lung tissues, alveolar septal thickness, alveolar bleeding and alveolar fibrin infiltration degree were used to determine the severity of lung pathology. Each item was scored on a five-point scale ranging from 0 to 4: No damage, mild damage, moderate damage, severe damage, and extremely severe damage are represented by 0, l, 2, 3, and 4, respectively, as described[42]. All samples were measured in at least five random fields. The average value under each field was taken as the sample value. For quantification of neutrophils in lung, two to three random fields per sample were performed.

## Flow cytometry

Blood cells were collected from the mouse tail. For surface staining, cells were washed once in FACS buffer (0.5% BSA in PBS), blocked with normal mouse serum for 10 min at 4 °C and incubated with the indicated antibodies for 30 min at 4 °C. Then, the cells were washed with FACS buffer and fixed in FACS buffer containing 1% PFA. For intracellular staining, cells were fixed and permeabilized with BD Fixation/Permeabilization solution (BD Biosciences, 554722) for 10 min at 4 °C. Then, the cells were washed with BD Perm/Wash™ buffer (BD Biosciences, 554723) and incubated with FITC conjugated PRL2 antibody for 30 min at 4 °C. The fluorescence-labeled PRL2 antibody was prepared as previously described[18]. Finally, the cells were washed with BD Perm/Wash™ buffer. Flow cytometric data were acquired on a BD LSRFortessa flow cytometer and analyzed by FlowJo software. The antibodies used were as follows: rat anti-mouse CD3e APC antibody (1:100, eBioscience, 17-0032-82, 17A2), rat anti-mouse/human B220 Brilliant Violet 421 antibody (1:100, BioLegend, 103239, RA3-6B2), rat anti-mouse CD11b PE antibody (1:100, eBioscience, 12-0112-83, M1/70), rat anti-mouse Ly-6G Brilliant Violet 510 antibody (1:100, BioLegend, 127633, 1A8) and rat anti-mouse Ly-6C PerCP/Cyanine5.5 antibody (1:100, BioLegend, 128011, HK1.4).

## Serum inflammatory cytokine screening

Serum was collected from mice which infected with PbA for 7 days. Concentrations of CCL-2, GM-CSF, IFN-β, IFN-γ, IL-1α, IL-6, IL-10, and TNF-α were determined by using the LEGENDplex Mouse Inflammation Panel (BioLegend, 740446) according to the manufacturer's instructions. In brief, 25 µl of diluted samples or standards, 25 µl of assay buffer or Matrix C and 25 µl of mixed bead solution were incubated together for 2 h at room temperature (RT) with optimal shaking in the dark. The plate was washed once and 25 µl of detection antibodies was added to each well for 1 h with shaking at RT in the dark. Then, 25 µl of SA-PE was added to each well directly and shaken for 30 min. After washing once, the samples were suspended in 200 µl of 1× wash buffer and collected by a BD LSRFortessa flow cytometer. Data were analyzed by LEGENDplex™ Data Analysis Software and calculated by standard curves according to the manufacturer's instructions.

## Measurement of NETs ex vivo

Neutrophil extracellular traps (NETs) were counted in mouse blood samples by conventional Giemsa microscopy or immunofluorescence.

In Giemsa-stained thin smears, NETs were identified under a 100× oil objective lens as extracellular structures with staining of decondensed chromatin associated with fragmented neutrophil-like cells or small granules[12].

For immunofluorescence, blood cells were collected, washed once in FACS buffer, fixed and permeabilized with BD Fixation/Permeabilization solution for 10 min at 4 °C. Then, the cells were washed with BD Perm/Wash™ buffer and incubated with MPO antibody (1:200, R&D, AF3667) and citrullinated histone H3 antibody (1:200, Abcam, ab5103) overnight at 4 °C. Cells were washed once in FACS buffer. The secondary antibodies used were Alexa Fluor 488 conjugated anti-goat IgG (1:1000, Abcam, ab150129) and Alexa Fluor 555 conjugated anti-rabbit IgG (1:1000, CST, 4413). After staining for 1 h at RT, samples were washed once in FACS buffer and stained with DAPI Fluoromount-G™ (Yeasen, 36308ES20). Images were obtained by laser confocal microscope (Leica TCS SP8) using a 63× oil objective lens. For quantification, at least five random fields per sample were analyzed. The average value under each field was taken as the sample value.

## MPO-DNA ELISA

Serum was collected from mice infected with PbA for 7 days. Biotinylated MPO antibody (Hycult, HBT-HM1051BT) was added to a streptavidin coated plate from the Cell Death Detection ELISA Kit (Roche, 11920685001) at 4 °C overnight, followed by three washes with PBST (0.05% Tween-20 in PBS). The plates were subsequently blocked for 2 h with 1% BSA in PBS at RT. Following three washes, 50 µl of mouse serum was added to each well and incubated for 2 h at RT with shaking. Then, 50 µl of anti-DNA-POD from the Cell Death Detection ELISA Kit was added to each well, incubated for 5 h at RT and washed with PBST for five times. Another 100 µl of ABTS was added. After incubation for 10-20 min at RT, 100 µl of stop solution was added. The optical density was measured at 405 nm using a microplate reader (Biotek, USA).

## Immunofluorescence of fixed tissues

Lungs were fixed in 4% PFA for 48 h, washed with PBS, embedded in paraffin and sectioned at 4 µm thicknesses. After deparaffinization and antigen retrieval, samples were permeabilized with 0.2% Triton X-100 in PBS for 15 min at RT. Then, the samples were blocked with 5% BSA in PBS for 1 h at RT. Subsequently, the samples were incubated with MPO antibody and citrullinated histone H3 antibody overnight at 4 °C. The secondary antibodies used were Alexa Fluor 488 conjugated anti-goat IgG and Alexa Fluor 555 conjugated anti-rabbit IgG. After staining for 1 h at RT, samples were stained with DAPI Fluoromount-G™. Images were obtained by laser confocal microscope using a 63× oil objective lens. For quantification, at least three random fields per sample were performed.

## Intravital multiphoton microscopy

Intravital imaging of the lung was performed as described previously[43]. In brief, six- to eight-week old female mice were anesthetized and placed on a heating pad. One microgram of rat anti-mouse Ly-6G PE antibody (BioLegend, 127607, 1A8) and 5 nmol of the noncell permeable dsDNA dye Sytox Green (Invitrogen, S7020) were delivered via vein injection per mouse. The mouse was connected to a small rodent respirator, the lung was exposed and stabilized with a small suction chamber. Images were observed using a multiphoton microscope with two tunable lasers (Olympus, FVMPE-RS-TWIN) and were taken every 10 s by collecting z-stacks of approximately 6 μm, with a 3 μm step size for 30 min.

## Purification of infected RBCs

Six- to eight-week old mice were intraperitoneally injected with $1 \times 10^6$ PbA iRBCs. When the percentage of iRBCs in peripheral blood reached 10%, blood was collected and washed twice with Hank's Balanced Salt Solution (HBSS) by centrifugating at $500 \times g$ for 5 min. Then, the blood was layered on a 2-step Percoll gradient (65%, 35%), and centrifuged at $1500 \times g$ for 15 min at RT without braking. IRBCs were enriched on a Percoll step gradient consisting of an upper 35% and a lower 65% Percoll layer. The iRBCs were collected, washed twice and diluted in HBSS. Purity was >90% as determined by Giemsa staining.

## Primary neutrophil isolation

Mouse neutrophils were purified from bone marrow by Percoll gradient centrifugation as previously described with some modifications[44]. Briefly, bone marrow cells were harvested from mice in ice-cold HBSS (without $Ca^{2+}$ and $Mg^{2+}$ containing 0.5% BSA). After centrifugation at $500 \times g$ for 5 min at 4 °C, the cells were layered on a 3-step Percoll gradient (78%, 65%, 55%), and centrifuged at $800 \times g$ for 20 min at RT without braking. Cells at the 78%−65% interface were collected and washed. After purification, neutrophil viability was >95%, as detected by trypan blue staining. Purity was typically >90%, as determined by morphological examination of Giemsa staining and flow cytometry based on cell surface staining of $CD11b^+$ and $Ly6G^+$.

## Visualization of NETs in vitro

Mouse neutrophils were allowed to rest for 1 h, pretreated with vehicle, 20 mM NAC (Sigma-Aldrich, A9165) or 50 mM NSC23766 (To CRIS, 2161) for 0.5 h, and then stimulated with 100 ng/ml PMA (Sigma-Aldrich, P1585), 20 μM hemin (Frontier, H651-9) and MOI = 1 iRBCs purified as described above for 4−6 h. An equal volume of PBS or DMSO was added as a control. Then, the cells were stained with 0.5 μM Sytox Green for 10 min. Following remove of the supernatant and washing twice, the cells were fixed with 4% PFA in PBS for 15 min at RT. After another wash, the cells were stained with DAPI Fluoromount-G™ and images were obtained with a Leica SP8 confocal microscope using a 63× oil objective lens. For quantification, NETs were counted in at least five random fields per sample.

## Live cell imaging

For live monitoring of NETosis events in vitro, mouse bone marrow neutrophils (BMNs) were isolated as described above. Neutrophils were seeded in glass bottom dishes (Jet biofil, BDD012035-N), rested for 1 h and stimulated with indicated reagents. Just before imaging, 0.5 μM Sytox Green was added to the medium to allow visualization of NETs. Images were obtained with a Leica SP8 confocal microscope using a 20× dry objective lens. Images were captured every 1.5 min for 30 min at 37 °C. For quantification, at least nine independent dishes were observed.

## Detection of ROS

ROS production was measured by luminol dependent chemiluminescence assay. Neutrophils ($1 \times 10^5$) were plated in a 96-well luminometer plate (Coster, 3917) and rested for 30 min. Prewarmed PBS or 100 ng/ml PMA was added together with 20 μM luminol (Sigma-Aldrich, v900354) and 20 μg/ml HRP (Sigma-Aldrich, OR03L). Chemiluminescence was measured immediately at 2.5 min intervals for 120 min with a luminometer (BioTek Synergy HT microplate reader).

ROS production was also detected by using CM-H$_2$DCFDA (ThermoFisher, C6827). After stimulation with PBS, 100 ng/ml PMA, 20 μM hemin and MOI = 1 iRBCs for 2 h, neutrophils were incubated with the fluorogenic probe CM-H$_2$DCFDA for 30 min at 37 °C in 5% $CO_2$. ROS were determined by a BD LSRFortessa flow cytometer and analyzed by FlowJo software.

## PAK pulldown and western blot

To measure Rac activity, neutrophils were isolated as described above, stimulated with 100 ng/ml PMA for 0, 5, 15 or 30 min and then washed twice with ice-cold HBSS. A total of $2 \times 10^6$ cells were lysed in 500 μl ice-cold lysis buffer (50 mM Tris-HCl, pH 8, 150 mM NaCl, 10 mM MgCl$_2$, 1 mM EDTA, 1% Triton X-100, 1 mM PMSF and 1× protease inhibitor cocktail). Then, 10 μl 5×SDS-PAGE sample buffer was added to 40 μl lysate and boiled for 5 min. The remaining lysates were incubated with 20 μg PAK-GST protein beads (Cytoskeleton, PAK02) for 30 min at 4 °C. Beads were washed three times with lysis buffer, resuspended in 20 μl 2×SDS- PAGE sample buffer and boiled. Both the lysis and bead samples were separated by SDS-PAGE, transferred to a nitrocellulose membrane, blocked with 5% (w/v) nonfat dried milk in TBST and blotted with specific antibodies. The membrane was developed using Thermo SuperSignal reagent (ThermoFisher, 34577) and detected by ImageQuant LAS 4000 mini (GE, USA). Antibodies were used as follows: PRL2 antibody (1:1000, Millipore, 05-1583, clone 42), GAPDH antibody (1:1000, Proteintech, 10494-1-AP), Rac1/2/3 antibody (1:1000, CST, 2465), HRP linked anti mouse IgG (1:2000, CST, 7076) and HRP linked anti rabbit IgG (1:2000, CST, 7074).

## Rac GTPase activity assay

BMNs were isolated as described above, rested for 1 h in HBSS containing 0.5% BSA and stimulated with 100 ng/ml PMA for 5 min. After stimulation, cells were washed with ice-cold PBS and fixed with 4% PFA for 15 min at RT. Then, the cells were permeabilized in PBS containing 0.1% Triton X-100 and 3% BSA for 10 min at RT and blocked with PBS containing 5% normal goat serum and 3% BSA for 1 h at RT. Cells were stained overnight at 4 °C with Rac-GTP antibody (1:100, NewEast Biosciences, 26903) in 3% BSA, then incubated with Alexa Fluor 488 conjugated anti-mouse IgG/IgM (1:1000, ThermoFisher, A-10680) in 3% BSA for 1 h at RT. Finally, the cells were stained with DAPI Fluoromount-G™. Images were obtained with a Leica SP8 confocal microscope using a 63× oil objective lens and analyzed by ImageJ software. For quantification, more than 25 cells of each type and condition were analyzed.

## Statistical analyses

Data are presented as the mean ± standard error of mean (SEM) or in violin plots showing the median and interquartile range. Data were analyzed by two-tailed unpaired t test, two-tailed Mann−Whitney test or ANOVA according to the experiments using GraphPad Prism software. Survival analysis was performed according to the Kaplan−Meier method and the log-rank test. Spearman's correlation analysis was used to analyze the association between PRL2 levels and clinical scores. A $p$ value less than 0.05 was considered statistically significant.

## Reporting summary

Further information on research design is available in the Nature Portfolio Reporting Summary linked to this article.

## Data availability

All data associated with this study are available in the article and the supplementary information. The source data are provided as a Source Data file. Uncropped images are available at FigShare (https://doi.org/10.6084/m9.figshare.23896062). Source data are provided with this paper.

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

## Acknowledgements

We thank Prof. Ivo Mueller of the University of Melbourne and Prof. Xiaonong Zhou of Chinese Center for Tropical Diseases Research for

their advice. This work was supported by the National Natural Science Foundation of China (81971486 and 82272362). S. C. W. is supported by the National Institute of Allergy and Infectious Diseases of the National Institutes of Health (NIH) under award number U19AI089676, and the Medical Research Council UK under award number MR/S009450/1. The content is solely the responsibility of the authors and does not necessarily represent the official views of the NIH.

## Author contributions

Z.W. generated the initial ideas and proposed the hypotheses. Z.W., X.D., B.R. and J.W. designed the study. Z.W., G.C. and M.Y. supervised the study. X.D. and B.R. conducted the key experiments. C.L., Q.L., S.K., X.W., C.W., H.Y. and X.N. performed experiments. J.W. provided the platform of intravital two-photon microscopy and W.B. performed related experiments. Z.W., X.D., B.R., W.X. and G.C. performed the data analysis and interpretation of the results. Z.W., X.D., S.W. and K.K. wrote the manuscript.

## Competing interests

The authors declare no competing interests.
