## [Peer Review File · Nature Communications]

PRL2 regulates neutrophil extracellular trap formation which contributes to severe malaria and acute lung injuryREVIEWER COMMENTS

Reviewer #1 (Remarks to the Author):

I have gone through the manuscript titled "PRL2 regulates neutrophil extracellular trap formation contributing to severe malaria and acute lung injury" by Du, X et al. Authors have performed extensive study to show co-relation of Phosphatase of regenerating liver 2 (PRL2) (A protein tyrosine phosphatase family protein) with lung injury during severe malaria. Although authors have performed extensive series of experiments to prove their results, I still feel that two important aspects are missing in this manuscript. (1) Authors have not identified a parasite protein that is responsible for generating PRL2 mediated immunity during infection. What happens to this immunity during normal malaria diseases. (2) Why PRL2 immunity is only linked to lung injury in severe malaria. In spite of these queries, I still recommend the publication of this manuscript after some revisions. My queries are

1. What kind of clinical scores authors took into considerations. (Fig 1A). Please write in detail.
2. How many mice were taken in each group and how many died during course of infection? Please write number in result section, Page 5.
3. Lines 106- 115). How many times these experiments were repeated. Please give figures for each experiment in supplementary section.
4. Images showing merger of staining (Ly6G and Sytox Green (SG) in CKO mouse are not clear (Figure 2). How many times and how many images were taken? Many of these images should be represented in supplementary figure
5. Figure 3A, Was any marker used for NETosis. Otherwise it should be marked by arrows
- 6 Figure 3B. Hemen does not cause dramatic increase in NET formation. any explanation. It should cause more drastic effect than iRBC. It suggest that some parasite surface porotein promotes NET formation. Did authors try to identify any of the iRBC surface protein
7. How the link between NET formation and ROS generation was established?

Reviewer #2 (Remarks to the Author):

Du, Ren et al present interesting data on PRL2 and its possible role in controlling lung related manifestations of malaria in a murine model. Levels of PRL2 are decreased in mice with severe malaria, especially in myeloid cells. Conditional knock out of PRL2 seems to increase neutrophil accumulation in the lungs of the mice, and some limited data on possible effects on survival are presented. The dissection of the pathways by which PRL2 is more interesting than the disease phenotype, but it does veer away from the initial focus on malaria, especially the LPS model, which is pathologically quite clearly more aggressive than the initial malaria model. Given that mice did not have increased TNF, the TNF priming model may also not be very relevant. The authors do not make a compelling case that this pathway is paying a major role in malaria pathogenesis, and without this or equivalent hman data the significance may be limited

Major comments

1. The most striking finding in the pulmonary pathology, relative to published work (Sercundes et al PLoS Pathogens 2016), is the apparent lack of alveolar exudate in the lungs of the mice (although there is certainly impressive accumulation of neutrophils in alveolar vessels, as the data show). More impressive changes are seen in the lungs of LPS treated mice. What is the evidence that the intravascular neutrophil accumulation is actually causing serious effects in the mice?
2. Related to this, in Figure 1 a (fourth panel!) only 33% of infected mice survive challenge to 14 days, but in 1 f (where the CKO mice are compared) 3 of 6 control mice do. These are the only in vivo data we have in the mice to understand the possible severity of the condition under study or the in vivo effect of the knock out on outcomes.
3. The authors are very parsimonious with figure labelling. Often multiple images and accompanying graphs are covered by a single letter and it can be very difficult to follow which image relates to which data. This also means that key data such as pulmonary histology are reduced to postage stamp sized images which are very hard to interpret. It is not clear for

example that there is erythrocyte extravasation into the alveolar spaces in Figure 1h. (see also comment 1 for the relevance of this).

4. The HCQ effect in the TNF and LPS models is clear, But does it have an effect in the disease survival? And how does one untangle anti-parasitic from anti-inflammatory effects?

5. Methods: More details on the *P. berghei* strain used would be helpful. These vary quite substantially in their pathogenicity. The ANKA strain for example usually causes cerebral malaria in C57/Bl6 mice. Given that the scoring system used for severity is based on cerebral symptoms, this is important to clarify.

6. Methods" What is MPO-DNA (Figure 1 I) and how is it measured?

Minor comments

1. Introduction. The respiratory distress syndrome that is the third major component of child deaths from malaria is not ARDS/ALI, which is largely restricted to adults. It relates mostly to metabolic acidosis (see e.g. Mitran et al, *Microorganisms* 2023). See also line 128.

2. Line 55: "defend the host"?

3. Line 67-8 "in contrast, those of circulating lymphocytes usually decrease"

4. Line 72 "could serve"

5. Line 282 "severe"

6. Line 284-6 please reword sentence for clarity

Supplemental material

1. Charts are labelled as showing "standard error of measurement" (S F1). Here I think the authors mean standard error of the mean. For parts 3 and 4, it is not clear that this is what is shown, and a consistent style of display would be better. Given data are not clearly normal, median and IQR would be more appropriate. Are the centre lines medians or means?

2. In S F 3 it is not clear that there is any meaningful citrullinated H3 in the KO mice immunofluorescence. It is also not clear how a "neutrophil" differs from a "NET" in this figure. What's meant by "data were pooled from three independent experiments". Were each of these nine mice?

3. Supplementary movie 2 title: Please reword this, it is not clear what the figure legend means. Spell out BMNs and any other relevant abbreviations

Reviewer #3 (Remarks to the Author):

Malaria is an important disease in which the innate immune contribution to pathology is poorly understood. Similarly, our understanding of molecular regulation of neutrophils is very restricted. The manuscript by Du et al addresses both of these knowledge gaps.

However, the study uses experimental cerebral malaria (ECM) with *P. berghei*, which has been heavily criticized:

<https://www.ncbi.nlm.nih.gov/pmc/articles/PMC2807032/>

While mouse models can be a useful tool, attempts must be made to validate findings in humans, otherwise impact is greatly limited. The authors cannot use exclusively use ECM to make claims about severe malaria.

Furthermore the authors use a very non-specific myeloid conditional KO driver: *LysMCre+* This promoter is active in all myeloid cells, including microglia so cannot be used to make inferences about neutrophils.

Finally, the study is similar to previous findings by the same group showing that PRL2 regulates ROS via Rac signaling. Since ROS is a major regulator of NETs, it is unsurprising that PRL2 also regulates NETs.

Because of these drawbacks, and the methodological issues outlined below, my enthusiasm for this manuscript is reduced.

Specific comments:

The consensus in the field is that terms such as 'malaria' and 'severe malaria' cannot be applied to *P. berghei* ANKA. These should be replaced with ECM.

Figure 1: *Pb* ANKA is a standard model, panels A and B are redundant with multiple reports in the literature and can be moved to supplement.

Lines 114: Further analysis showed that mice with severe malaria had significantly decreased protein levels of PRL2 compared to mice with uncomplicated disease (Figure 1E).

How is 'severe malaria' defined here? P

Figure 1F: there is currently no good evidence that neutrophils contribute to death in the *Pb* anka mouse model. Old studies used the anti-GR1 antibody to deplete neutrophils and found an effect on survival, however this has not been replicated with the more specific anti-Ly6G. In order to make the claim that PRL2 in neutrophils is promoting survival, the authors first need to demonstrate that specific neutrophil depletion is important for survival in this model. PRL2 may be important in monocytes or macrophages instead.

Figure 1K: detection of NETs with Giemsa is not specific and not appropriate. More reliable methods should be used, such as the ones employed in the lung histology later on.

1L: what is the y axis in this experiment? Is this relative?

Figure 2: these findings are interesting and the microscopy is well done. However there needs to be confirmation that elevated neutrophils/NETs are in fact contributing to ALI pathology in this model. Neutrophils/NETs must be depleted to demonstrate this.

Figure 3: there appears to be a major problem with this experiment. The control cells in both WT and cKO are green in the SG channel, indicating that the cells are dead. The negative control has failed - the authors are therefore working with dead neutrophils, making it difficult to interpret any of the other results.

Figure 4: this appears to already be published by the same authors in a previous report, therefore lacks novelty.

Figure 5: what is the advantage of setting up a more artificial model of malaria ALI in this figure? Please show the results of WT versus cKO in both models in the main figures:

this is the only piece of data on causation – correlation of PRL2 abundance is only an association.

Figure 6: the anti-inflammatory effects of chloroquine are well described in vitro and in mouse models. Its use in a new model is interesting but there is no formal proof that it exerts this anti-inflammatory effect via stabilization of PRL2. More robust experiments are needed to make that link

● Response to Reviewers

We thank all 3 reviewers for their critical evaluation of this manuscript. All comments are valuable and very helpful for revising and improving our manuscript, as well as the important guiding significance to our researches. This revision has taken 3 months because we performed additional experiments to carefully address all the issues and concerns raised by the reviewers, and make detailed revisions according to the suggestion.

In the revised manuscript, we have addressed the reviewers' concerns thoroughly. Below are our point-by-point responses to the reviewers' comments (shown in blue). The changes made to the text of the revised manuscript are highlight in yellow. Page numbers corresponding to these changes are provided in this response letter.

Reviewer #1:

I have gone through the manuscript titled "PRL2 regulates neutrophil extracellular trap formation contributing to severe malaria and acute lung injury" by Du, X et al. Authors have performed extensive study to show co-relation of Phosphatase of regenerating liver 2 (PRL2) (A protein tyrosine phosphatase family protein) with lung injury during severe malaria. Although authors have performed extensive series of experiments to prove their results, I still feel that two important aspects are missing in this manuscript. (1) Authors have not identified a parasite protein that is responsible for generating PRL2 mediated immunity during infection. What happens to this immunity during normal malaria diseases. (2) Why PRL2 immunity is only linked to lung injury in severe malaria. In spite of these queries, I still recommend the publication of this manuscript after some revisions.

Response:

We sincerely appreciate the reviewer's positive evaluation and helpful comments of this manuscript. (1) In this study, we found PRL2 is involved in parasite infected RBC (iRBC) and heme induced NET formation. However, the parasite protein responsible for this remains elusive and we have added a sentence to this effect in the Discussion, emphasizing that further investigations are warranted. (2) *P. berghei* infection caused more severe disease in PRL2 CKO mice than in WT control mice. We compared brain, lung, liver and kidney pathology or function between two groups (Figure 1 and new supplementary Figure 2). We have added a sentence to potentially explain why the PRL2 immunity is mainly linked to lung injury in severe malaria, hypothesizing that it is linked to the specific anatomical structure and microenvironment of lung tissues. It has been reported, NETs can expand more easily in the pulmonary alveoli. They are able to directly induce epithelial and endothelial cell death. Thus, the detrimental effect of excessive NET release is particularly important to lung injury¹. Above information is added on Page 12 in the Discussion of revised manuscript.

My queries are

1. What kind of clinical scores authors took into considerations. (Fig 1A). Please write in detail.

Response: We apologize if this was not clear enough. The signs are as follows: 0 is no signs, 1 to ruffled fur, 2 to hunching, 3 to wobbly gait, 4 to limb paralysis, 5 to convulsions, 6 to coma. And mice with scores greater than or equal to 3 were considered to have severe malaria. Above detail information has been described in method section on Page 15. We also put the detail information in the legends of new Figure 1d in the revised manuscript.

2. How many mice were taken in each group and how many died during course of infection? Please write number in result section, Page 5.

Response: To establish a mouse model of malaria, C57BL/6J mice (n=6 per group) were intraperitoneally inoculated with 1×10^6 PbA infected RBCs or PBS control. Mice began to show severe malaria symptoms on the 7th day post infection (dpi) and a portion of mice (4 in 6) died during the second week of PbA infection. The results are pooled data from two independent experiments, three pairs of mice per experiment. Above information has been added in the revised Results on Page 5 and the legends of new supplementary Figure 1.

3. Lines 106- 115). How many times these experiments were repeated. Please give figures for each experiment in supplementary section.

Response: The words (Lines 106 – 115) were associated with three original figures, original Figure 1c-e (new Figure 1a-c). These data are pooled from two independent experiments. As suggested, figures for each experiment have been provide in supplementary section (new supplementary Figure 1k-m).

4. Images showing merger of staining (Ly6G and Sytox Green (SG) in CKO mouse are not clear (Figure 2). How many times and how many images were taken? Many of these images should be represented in supplementary figure.

Response: The representative images in original Figure 2b were updated (new Figure 2c).

The experiments of intravital imaging were repeated twice, totaling 6 mice per group. One region was taken from each mouse. As suggested, time-lapse images and videos of another 5 mice have been shown in new Supplementary Figure 3 and new Supplementary Movie 2.

5. Figure 3A, was any marker used for NETosis. Otherwise, it should be marked by arrows.

Response: We appreciate the Reviewer's suggestions. Arrows were added in the images (new Figure 4a).

6. Figure 3B. Hemen does not cause dramatic increase in NET formation. any explanation. It should cause more drastic effect than iRBC. It suggests that some parasite surface porotein promotes NET formation. Did authors try to identify any of the iRBC surface protein.

Response: We greatly agree with the reviewer that hemin is a stronger NETs inducer. In our *in vitro* system, Hemin stimulation caused 34.4% and 46.2% NETs in WT and PRL2 KO neutrophils respectively. While iRBC stimulation caused 15.3% and 29.7% NETs in WT and PRL2 KO cells respectively (Figure 4b). As a purified component Hemin did induce more drastic effect than iRBCs. We appreciate the reviewer's suggestion. PRL2 is involved in both iRBCs and hemin induced NET formation. However, we still don't know which parasite protein might be responsible for it. It is an interesting question and worth to be further investigated. The associated information was added on Page12 and 13 in the Discussion of revised manuscript.

7. How the link between NET formation and ROS generation was established?

Response: Evidence suggests that ROS are the key events leading NET formation^{2,3}. ROS generated by NADPH oxidase stimulate myeloperoxidase (MPO) to trigger the activation and translocation of neutrophil elastase (NE) from neutrophil granules. Consequently, the migration of elastase into neutrophil nuclei leads to chromatin decondensation⁴. In our experiments, the antioxidant N-acetylcysteine (NAC) inhibited NET release and abrogated the difference between WT and PRL2 KO cells. It proves that PRL2 related NET formation is dependent on ROS. The associated information was provided on Page 13 in the Discussion.

Reviewer #2:

Du, Ren et al present interesting data on PRL2 and its possible role in controlling lung related manifestations of malaria in a murine model. Levels of PRL2 are decreased in mice with severe malaria, especially in myeloid cells. Conditional knock out of PRL2 seems to increase neutrophil accumulation in the lungs of the mice, and some limited data on possible effects on survival are presented. The dissection of the pathways by which PRL2 is more interesting than the disease phenotype, but it does veer away from the initial focus on malaria, especially the LPS model, which is pathologically quite clearly more aggressive than the initial malaria model. Given that mice did not have increased TNF, the TNF priming model may also not be very relevant. The authors do not make a compelling case that this pathway is playing a major role in malaria pathogenesis, and without this or equivalent human data the significance may be limited.

Response: We thank the reviewer for their constructive criticism. Although LPS model is not associated with malaria pathogenesis per se, this model supports the importance of PRL2 in NET formation and tissue injury. Our finding may be relevant to other NET-associated pathologies beside malaria infection. To avoid confusion, we moved all LPS model data to the supplementary section.

TNF is an important pathogenic cytokine in malaria, as demonstrated by a systematic review and meta-analysis of 31 studies carried out in 2022 by Mahittikorn et al ⁵. While cytokines are produced in waves and TNF levels are fluctuating during infection, they confirmed the overall increase of TNF levels in patients with severe malaria. TNF is also considered as an important pathogenic factor in the malaria mouse model which is associated with ARDS ^{6,7}. In our *in vivo* system, mice with PbA infection did show higher TNF levels compared to those without infection (new Supplementary Figure 1j). We therefore believe that our TNF-priming model is relevant to malaria infection, especially the pathogenetic mechanisms underlying the development of severe malaria.

Malaria-associated acute respiratory distress syndrome (MA-ARDS) and acute lung injury (ALI) are complications that cause lung damage and often leads to death. ARDS is probably the most extreme form of a continuum of pulmonary involvement in malaria, ranging from highly prevalent mild respiratory symptoms (e.g., coughing) to ALI and ARDS ⁸. As reported, malaria-associated (MA)-ARDS occurs mainly in adults and often leads to rapid deterioration and a poor prognosis with lethality rates up to 80% ⁹. MA-ARDS is the most prevalent complication of *Plasmodium knowlesi* infections in patients (59-70% of severe cases) and affects 5-25% adults infected with *Plasmodium falciparum* ^{9,10}. Our study suggests the important role of PRL2 in NET formation and pathogenesis of ALI during infections by all these different species of malaria parasites. We agree that our study has limitations and have mentioned the need for result validation in patients at the end of the discussion on Page 14.

Major comments

1. The most striking finding in the pulmonary pathology, relative to published work (Sercundes et al PLoS Pathogens 2016), is the apparent lack of alveolar exudate in the lungs of the mice (although there is certainly impressive accumulation of neutrophils in alveolar vessels, as the data show). More impressive changes are seen in the lungs of LPS treated mice. What is the evidence that the intravascular neutrophil accumulation is actually causing serious effects in the mice?

Response: To address this question, we used anti-Ly6G antibody to selectively deplete neutrophils in WT and PRL2 CKO mice respectively. The results showed that the depletion of neutrophils when the first clinical signs appear and prior to the onset of severe disease significantly protected PRL2 CKO mice. Decreased inflammatory infiltration and NETs accumulation in the lung were observed in both WT and PRL2 CKO mice. Importantly, neutrophil depletion reduced the difference in pulmonary pathology between WT and CKO mice (new Figure 3). The information is discussed on Page 7-8. The methods have been updated accordingly on Page 16.

2. Related to this, in Figure 1 a (fourth panel!) only 33% of infected mice survive challenge to 14 days, but in 1 f (where the CKO mice are compared) 3 of 6 control mice do. These are the only *in vivo* data we have in the mice to understand the possible severity of the condition under study or the *in vivo* effect of the knock out on outcomes.

Response: Data in original Figure 1a (new Supplementary Figure 1a) are pooled data from two independent experiments, with three pairs of mice per experiment. Mice began to show severe malaria signs on the 7th day post infection (dpi) and a portion of mice (4 of 6) died during the second week of PbA infection.

Results in original Figure 1f (new Figure 1d) are pooled data from two other independent experiments, three pairs of mice per experiment. WT and CKO were infected with PbA at the same time in each independent experiment. All the CKO mice (6 of 6) died within nine days, whereas 50% of the WT mice (3 of 6) remained alive until fourteen days after PbA infection.

Besides the survival rate and histopathology of the lung tissue, we now have added new *in vivo* evidence. Computed Tomography (CT) was used to detect edema in PbA infected WT and PRL2 CKO mice. As shown in new Figure 1f, PRL2 CKO mice showed more obvious ground-glass opacification pulmonary lesions at 7 dpi. The information is discussed on Page 7. The methods have been updated accordingly on Page 15 and 16.

3. The authors are very parsimonious with figure labelling. Often multiple images

and accompanying graphs are covered by a single letter and it can be very difficult to follow which image relates to which data. This also means that key data such as pulmonary histology are reduced to postage stamp sized images which are very hard to interpret. It is not clear for example that there is erythrocyte extravasation into the alveolar spaces in Figure 1h. (see also comment 1 for the relevance of this).

Response: We apologize for these omissions. The figures are now clearly reorganized and labeled. All histological images have been enlarged and updated, including Figure 1h. Computed tomography (CT) data has been added to the results as described above.

4. The HCQ effect in the TNF and LPS models is clear, but does it have an effect in the disease survival? And how does one untangle anti-parasitic from anti-inflammatory effects?

Response: We sincerely thank for the reviewer's agreement with HCQ effect in the TNF- α and LPS model. HCQ has both anti-parasite and anti-inflammatory effects. Since PbA parasites generally take 48 hours to finish their replication cycle in RBCs, the 24-hour TNF priming iRBCs model may highlight more specifically the anti-inflammatory effects of HCQ. The results of LPS model also supported the anti-inflammatory effects of HCQ.

5. Methods: More details on the *P. berghei* strain used would be helpful. These vary quite substantially in their pathogenicity. The ANKA strain for example usually causes cerebral malaria in C57/Bl6 mice. Given that the scoring system used for severity is based on cerebral symptoms, this is important to clarify.

Response: As suggested, the strain name was added in the revised manuscript. The ANKA strain does cause experimental cerebral malaria in C57/Bl6 mice. Adherence of iRBCs and leucocytes to brain vessels and vascular plugging in both WT and CKO mice were observed. Blood-brain barrier (BBB) permeability was increased after infection (new supplementary Figure 1, 2), especially in WT mice at 10 dpi. We compared the brain injury between WT and CKO mice on the 7th dpi. There was no dramatic difference in the two groups. PRL2 CKO mice are more susceptible to severe malaria with lung injury and died before showing typical signs of experimental cerebral malaria.

The signs of disease development were scored daily as follows: 0 is no signs, 1 to ruffled fur, 2 to hunching, 3 to wobbly gait, 4 to limb paralysis, 5 to convulsions, 6 to coma. This scoring system was described in Methods (Page 15, line 321-323) and Figure legends 1d. These scores also present an overall status of the infected mice.

6. Methods" What is MPO-DNA (Figure 1 l) and how is it measured?

Response: NETs contain both DNA and myeloperoxidase (MPO). As reported, MPO–DNA complex ELISA can be used to detect and quantify soluble NETs in serum or culture supernatants¹¹. The detail information has been provided in the Methods section on Page 19.

Minor comments

1. Introduction. The respiratory distress syndrome that is the third major component of child deaths from malaria is not ARDS/ALI, which is largely restricted to adults. It relates mostly to metabolic acidosis (see e.g., Mitran et al, *Microorganisms* 2023). See also line 128.

Response: Our original words in introduction are “Severe malaria is mainly caused by *Plasmodium falciparum* in which cerebral malaria, severe anemia, and acute respiratory distress syndrome/acute lung injury (ARDS/ALI) are the three most common complications”. It was according to information collected from literatures. As described by Ashley et. al in *Lancet* (2018), “The most common manifestations of severe malaria are cerebral malaria, acute lung injury, which can progress to acute respiratory distress syndrome (in up to 25% of cases), acute kidney injury, typically presenting as acute tubular necrosis, and acidosis”¹²; Kotepui et al mentioned in *Scientific Reports* (2020), “Both mixed infection and *P. falciparum* mono-infection showed a similar trend of complications in which severe anemia, pulmonary failure, and renal impairment were the three most common complications found”¹³.

To avoid confusion, our words were updated as “Severe malaria is mainly caused by *Plasmodium falciparum* in which cerebral malaria, severe anemia, and acute respiratory distress syndrome/acute lung injury (ARDS/ALI) are common complications” (Page 3 in the Introduction of revised manuscript).

2. Line 55: “defend the host”?

Response: As suggested, the words “the host” have been added.

3. Line 67-8 “in contrast, those of circulating lymphocytes usually decrease”

Response: The words have been updated as suggested.

4. Line 72 “could serve”

Response: We have changed the words according to the suggestion.

5. Line 282 “severe”

Response: This error has been corrected.

6. Line 284-6 please reword sentence for clarity

Response: As suggested, we reworded the sentence.

Supplemental material

1. Charts are labelled as showing “standard error of measurement” (S F1). Here I think the authors mean standard error of the mean. For parts 3 and 4, it is not clear that this is what is shown, and a consistent style of display would be better. Given data are not clearly normal, median and IQR would be more appropriate. Are the center lines medians or means?

Response: We apologize for these. In this study, SEM means “standard error of mean”. Data are presented as the mean \pm SEM. In violin plots, the center lines are medians. As suggested, text and display style have been updated.

2. In S F 3 it is not clear that there are any meaningful citrullinated H3 in the KO mice immunofluorescence. It is also not clear how a “neutrophil” differs from a “NET” in this figure. What’s meant by “data were pooled from three independent experiments”. Were each of these nine mice?

Response: These images in original S F 3. The original experiments were repeated three times for totally 9 mice per group. Since the space issue, original S F 3 was removed from the revised manuscript. Updated results were integrated into neutrophil depletion experiments as isotype control groups (new Figure 6m).

3. Supplementary movie 2 title: Please reword this, it is not clear what the figure legend means. Spell out BMNs and any other relevant abbreviations

Response: As suggested, the movie title was reworded as “Live cell imaging shows more NETosis in PRL2 KO neutrophils”. BMNs and relevant abbreviations were spell out. We thank the reviewer for their critical reading of the manuscript.

Reviewer #3:

Malaria is an important disease in which the innate immune contribution to pathology is poorly understood. Similarly, our understanding of molecular regulation of neutrophils is very restricted. The manuscript by Du et al addresses both of these knowledge gaps. However, the study uses experimental cerebral malaria (ECM) with *P. berghei*, which has been heavily criticized: <https://www.ncbi.nlm.nih.gov/pmc/articles/PMC2807032/>. While mouse models can be a useful tool, attempts must be made to validate findings in humans, otherwise impact is greatly limited. The authors cannot use exclusively use ECM to make claims about severe malaria.

Response: We appreciated the reviewer for this critical evaluation of this manuscript. We used *Plasmodium berghei* ANKA to infect C57BL/6 mice and the mice developed ECM as well as lung, liver and kidney injury (new supplementary Figure 1). We agree that PbA infection can be used to study experimental cerebral malaria (ECM), but it is also a well-known murine model of severe malaria, associated with multiple organ dysfunction^{14,15}. In addition to the brain, highly vascularized organs such as the lungs and kidneys are especially affected during *Plasmodium berghei* ANKA infection¹⁶. Here, we found all the PRL2 CKO mice died within nine days, whereas 50% of the WT mice remained alive until fourteen days after PbA infection. PRL2 CKO mice appeared to die more quickly than WT mice. We demonstrate that in our model, a reduction of PRL2 in malaria aggravate disease severity, with a pivotal role in the development of acute lung injury. We agree that these findings must now be validated in patients, and have added a sentence to this effect in the Discussion on Page 14.

In patient, the common manifestations of severe malaria include cerebral malaria, acute lung injury, which can progress to acute respiratory distress syndrome (in up to 25% of cases), acute kidney injury, typically presenting as acute tubular necrosis, and acidosis¹². Malaria-associated (MA)-ARDS occurs mainly in adults and often leads to rapid deterioration and a poor prognosis with lethality rates up to 80%. Our findings highlight new potential pathogenic mechanisms in the development of this pathology.

Furthermore, the authors use a very non-specific myeloid conditional KO driver: LysMCre+ This promoter is active in all myeloid cells, including microglia so cannot be used to make inferences about neutrophils.

Response: We thank the reviewer for pointing out this important issue. We carried out additional experiments to address this issue. An anti-Ly6G antibody was used to selectively deplete neutrophils. PbA infected WT and PRL2 myeloid CKO mice received a single dose of anti-Ly6G monoclonal antibody or control antibody prior to the onset of severe disease (new Figure 3a). The depletion of neutrophils was achieved on the second day after antibody treatment (new Figure

3b). A single dose anti-Ly6G treatment on the 6th dpi didn't show significantly effects on parasitemia and anemia degree in PbA infected WT and CKO mice (new Supplementary Figure 4). However, neutrophil depletion at this late stage of the infection successfully increased survival rate of PRL2 CKO mice and abolished the difference between WT and CKO mice (new Figure 3c). Remarkably, anti-Ly6G treatment reduced lung pathology associated with neutrophil infiltration and NET accumulation, thereby abrogating the difference observed between WT and PRL2 CKO group (new Figure 3d-g). This late treatment coincided with the appearance of the first signs (Figure 1d), showing that the pathogenic mechanisms we describe can still be stopped and reversed in infected animals. The above information has been added in the revised manuscript on Page 7 and 8. Methods are also updated on Page 16.

Finally, the study is similar to previous findings by the same group showing that PRL2 regulates ROS via Rac signaling. Since ROS is a major regulator of NETs, it is unsurprising that PRL2 also regulates NETs. Because of these drawbacks, and the methodological issues outlined below, my enthusiasm for this manuscript is reduced.

Response: Our previous studies indicated that, during bacterial infection, PRL2 regulates ROS production in phagocytes, mainly in macrophages ¹⁷. This manuscript focuses on NET formation, a specific cellular behavior of neutrophils. Since NETs formation can also be ROS independent ¹⁸, our study for the first time revealed that PRL2 regulates neutrophil activation and NET accumulation via the Rac-ROS pathway. Thus, PRL2 might be targeted to prevent pathologic NET formation, and applied in severe malaria or ALI-associated therapy.

Specific comments:

The consensus in the field is that terms such as 'malaria' and 'severe malaria' cannot be applied to *P. berghei* ANKA. These should be replaced with ECM.

Response: *P. berghei* ANKA infection does cause ECM, but as stated above, it is also a well-known murine model of severe malaria, associated with multiple organ dysfunction ^{14,15}. We have defined the different groups using an arbitrary severity threshold described in the methods, and given the significant differences between our two groups, we respectfully disagree and believe that using ECM would not be suitable in this study.

Figure 1: Pb ANKA is a standard model, panels A and B are redundant with multiple reports in the literature and can be moved to supplement.

Response: As suggested, original panels A and B of Figure 1 have been removed from the manuscript to the supplementary materials (new supplementary Figure

1 a).

Lines 114: Further analysis showed that mice with severe malaria had significantly decreased protein levels of PRL2 compared to mice with uncomplicated disease (Figure 1E). How is 'severe malaria' defined here?

Response: As mentioned above, we used an arbitrary threshold of disease severity based on the clinical scores monitored daily. Mice with scores greater than or equal to 3 were defined as having severe malaria ¹⁹. The information is described on Page 15.

Figure 1F: there is currently no good evidence that neutrophils contribute to death in the Pb anka mouse model. Old studies used the anti-GR1 antibody to deplete neutrophils and found an effect on survival, however this has not been replicated with the more specific anti-Ly6G. In order to make the claim that PRL2 in neutrophils is promoting survival, the authors first need to demonstrate that specific neutrophil depletion is important for survival in this model. PRL2 may be important in monocytes or macrophages instead.

Response: In this study, we provided evidence of the crucial role of PRL2 in regulating neutrophils during severe malaria, as its absence leads to death in the PbA-infected murine model. We investigated the role of neutrophils on the infection outcome by using an anti-Ly6G antibody to selectively deplete neutrophils in WT and PRL2 CKO mice at 6 dpi. We demonstrated that neutrophil depletion at this late stage of the infection (i.e., when animals show clinical signs) successfully increased the survival rate of PRL2 CKO mice and abolished the previously observed pathological difference between WT and CKO mice. Host-targeted adjunct therapies are needed to improve survival in severe malaria, and our findings suggest that inhibiting specific neutrophil pathways to decrease PRL2 and NET accumulation can be beneficial. Provided these findings are confirmed in humans, HCQ could be used to stop and reverse neutrophil-associated pathological mechanisms, and "buy time" while conventional anti-Plasmodium drugs eliminate the parasite.

Figure 1K: detection of NETs with Giemsa is not specific and not appropriate. More reliable methods should be used, such as the ones employed in the lung histology later on.

Response: We would like to respectfully point out that the detection of NETs using Giemsa has been previously mentioned in the literature ^{20,21}. However, to further address this issue, we used MPO and cit-H3 staining to detect NETs in mice blood. The results are consistent with previous observation (new Figure 11, m Supplementary Figure 2f). Method was added on Page 18.

1L: what is the y axis in this experiment? Is this relative?

Response: We are sorry for the confusion. The Y axis in this experiment is relative OD value of MPO-DNA complex level. Since NETs contain both DNA and myeloperoxidase (MPO), an MPO-DNA complex ELISA was used to detect and quantify soluble NETs in serum or culture supernatants as previously described ¹¹. NET ELISA in method section was renamed as MPO-DNA ELISA for clarity (Page 19).

Figure 2: these findings are interesting and the microscopy is well done. However there needs to be confirmation that elevated neutrophils/NETs are in fact contributing to ALI pathology in this model. Neutrophils/NETs must be depleted to demonstrate this.

Response: As mentioned above, we depleted neutrophils using anti-Ly6G on the 6th dpi. The new data did support neutrophils contributing to ALI pathology in this model.

Figure 3: there appears to be a major problem with this experiment. The control cells in both WT and cKO are green in the SG channel, indicating that the cells are dead. The negative control has failed - the authors are therefore working with dead neutrophils, making it difficult to interpret any of the other results.

Response: We thank the reviewer for their constructive comment. Sytox Green is a cell-impermeant DNA dye, but in the experiments, cells were fixed with 4% PFA after certain stimulation, and then stained with Sytox Green. The negative control may display green fluorescence while they have normal morphology. Above method has been described in literatures, similar phenomenon was observed in their negative controls ²²⁻²⁵. To avoid confusion, we repeated the experiment using unfixed cells and got the consistent results (new Supplementary Figure 5).

Figure 4: this appears to already be published by the same authors in a previous report, therefore lacks novelty.

Response: In this study, we focus on NET formation, a specific cellular behavior of neutrophils. Neutrophils use several different ways to produce and release NETs, depending on the stimulus used ¹⁸. Some of the ways required ROS, while others produced NETs without the need for it. It is the first time to describe the role of PRL2 in the formation of NET via ROS, and also the role of PRL2 related NET in malaria. This stimulation system is different than previously described. PRL2-ROS-NET axis is also not a predictable result.

Figure 5: what is the advantage of setting up a more artificial model of malaria ALI in this figure?

Response: Malaria disease can be due to direct effects of parasite proliferation as well as damage due to parasite infection induced immunopathology. Generally malarial parasites take 48 hours to complete their asexual multiplication in erythrocytes. 24-h malaria ALI models are more suitable to investigate the role of neutrophils in immunopathology.

Please show the results of WT versus cKO in both models in the main figures: this is the only piece of data on causation – correlation of PRL2 abundance is only an association.

Response: As suggested, WT versus cKO in iRBCs model have been shown in the main figures (new Figure 6).

Figure 6: the anti-inflammatory effects of chloroquine are well described in vitro and in mouse models. Its use in a new model is interesting but there is no formal proof that it exerts this anti-inflammatory effect via stabilization of PRL2. More robust experiments are needed to make that link.

Response: As suggested, we performed additional experiments. We used H₂O₂ in vitro to mimic the inflammatory environment. As shown in new supplementary figure 7, PRL2 levels in neutrophils were reduced when exposed to H₂O₂, while this effect can be abolished by HCQ pre-treatment.

References:

- (1) Porto, B. N.; Stein, R. T. Neutrophil Extracellular Traps in Pulmonary Diseases: Too Much of a Good Thing? *Frontiers in Immunology* **2016**, *7*.
- (2) Papayannopoulos, V. Neutrophil extracellular traps in immunity and disease. *Nature reviews. Immunology* **2018**, *18*, 134–147.
- (3) Chamardani, T. M.; Amiritavassoli, S. Inhibition of NETosis for treatment purposes: friend or foe? *Mol Cell Biochem* **2022**, *477*, 673–688.
- (4) Metzler, Kathleen D.; Goosmann, C.; Lubojemska, A.; Zychlinsky, A.; Papayannopoulos, V. A Myeloperoxidase-Containing Complex Regulates Neutrophil Elastase Release and Actin Dynamics during NETosis. *Cell Reports* **2014**, *8*, 883–896.
- (5) Mahittikorn, A.; Mala, W.; Srisuphanunt, M.; Masangkay, F. R.; Kotepui, K. U.; Wilairatana, P.; Kotepui, M. Tumour necrosis factor- α as a prognostic biomarker of severe malaria: a systematic review and meta-analysis. *Journal of Travel Medicine* **2022**, *29*.
- (6) Galvão-Filho, B.; de Castro, J. T.; Figueiredo, M. M.; Rosmaninho, C. G.; Antonelli, L.; Gazzinelli, R. T. The emergence of pathogenic TNF/iNOS producing dendritic cells (Tip-DCs) in a malaria model of acute respiratory distress syndrome

(ARDS) is dependent on CCR4. *Mucosal Immunol* **2019**, *12*, 312–322.

(7) Hernandez-Valladares, M.; Naessens, J.; Musoke, A. J.; Sekikawa, K.; Rihet, P.; ole-MoiYoi, O. K.; Busher, P.; Iraqi, F. A. Pathology of Tnf-deficient mice infected with Plasmodium chabaudi adami 408XZ. *Experimental Parasitology* **2006**, *114*, 271–278.

(8) Maguire, G. P.; Handojo, T.; Pain, M. C. F.; Kenangalem, E.; Price, R. N.; Tjitra, E.; Anstey, N. M. Lung Injury in Uncomplicated and Severe Falciparum Malaria: A Longitudinal Study in Papua, Indonesia. *The Journal of Infectious Diseases* **2005**, *192*, 1966–1974.

(9) Van den Steen, P. E.; Deroost, K.; Deckers, J.; Van Herck, E.; Struyf, S.; Opdenakker, G. Pathogenesis of malaria-associated acute respiratory distress syndrome. *Trends in Parasitology* **2013**, *29*, 346–358.

(10) Taylor, W. R. J.; Hanson, J.; Turner, G. D. H.; White, N. J.; Dondorp, A. M. Respiratory manifestations of malaria. *Chest* **2012**, *142*, 492–505.

(11) Caudrillier, A.; Kessenbrock, K.; Gilliss, B. M.; Nguyen, J. X.; Marques, M. B.; Monestier, M.; Toy, P.; Werb, Z.; Looney, M. R. Platelets induce neutrophil extracellular traps in transfusion-related acute lung injury. *J Clin Invest* **2012**, *122*, 2661–2671.

(12) Ashley, E. A.; Pyae Phyo, A.; Woodrow, C. J. Malaria. *The Lancet* **2018**, *391*, 1608–1621.

(13) Kotepui, M.; Kotepui, K. U.; De Jesus Milanez, G.; Masangkay, F. R. Plasmodium spp. mixed infection leading to severe malaria: a systematic review and meta-analysis. *Sci Rep* **2020**, *10*, 11068.

(14) Silva, L. S.; Peruchetti, D. B.; Silva-Aguiar, R. P.; Abreu, T. P.; Dal-Cheri, B. K. A.; Takiya, C. M.; Souza, M. C.; Henriques, M. G.; Pinheiro, A. A. S.; Caruso-Neves, C. The angiotensin II/AT1 receptor pathway mediates malaria-induced acute kidney injury. *PLoS One* **2018**, *13*, e0203836.

(15) Frevert, U.; Nacer, A.; Cabrera, M.; Movila, A.; Leberl, M. Imaging Plasmodium immunobiology in the liver, brain, and lung. *Parasitol Int* **2014**, *63*, 171–186.

(16) Souza, M. C.; Silva, J. D.; Pádua, T. A.; Torres, N. D.; Antunes, M. A.; Xisto, D. G.; Abreu, T. P.; Capelozzi, V. L.; Morales, M. M.; AA, S. P.; Caruso-Neves, C.; Henriques, M. G.; Rocco, P. R. Mesenchymal stromal cell therapy attenuated lung and kidney injury but not brain damage in experimental cerebral malaria. *Stem Cell Res Ther* **2015**, *6*, 102.

(17) Yin, C.; Wu, C.; Du, X.; Fang, Y.; Pu, J.; Wu, J.; Tang, L.; Zhao, W.; Weng, Y.; Guo, X.; Chen, G.; Wang, Z. PRL2 Controls Phagocyte Bactericidal Activity by Sensing and Regulating ROS. *Frontiers in Immunology* **2018**, *9*, 2609.

(18) Kenny, E. F.; Herzig, A.; Krüger, R.; Muth, A.; Mondal, S.; Thompson, P. R.; Brinkmann, V.; Bernuth, H. V.; Zychlinsky, A. Diverse stimuli engage different neutrophil extracellular trap pathways. *Elife* **2017**, *6*.

(19) Amante, F. H.; Engwerda, C. R.; Good, M. F. Experimental asexual blood stage malaria immunity. *Curr Protoc Immunol* **2011**, *Chapter 19*, Unit 19.14.

(20) Rocha, B. C.; Marques, P. E.; Leoratti, F. M. S.; Junqueira, C.; Pereira,

D. B.; Antonelli, L.; Menezes, G. B.; Golenbock, D. T.; Gazzinelli, R. T. Type I Interferon Transcriptional Signature in Neutrophils and Low-Density Granulocytes Are Associated with Tissue Damage in Malaria. *Cell Rep* **2015**, *13*, 2829–2841.

(21) Beiter, T.; Fragasso, A.; Hudemann, J.; Schild, M.; Steinacker, J.; Mooren, F. C.; Niess, A. M. Neutrophils release extracellular DNA traps in response to exercise. *Journal of Applied Physiology* **2014**, *117*, 325–333.

(22) Behnen, M.; Möller, S.; Brozek, A.; Klinger, M.; Laskay, T. Extracellular Acidification Inhibits the ROS-Dependent Formation of Neutrophil Extracellular Traps. *Front Immunol* **2017**, *8*, 184.

(23) Chauhan, D.; Demon, D.; Vande Walle, L.; Paerewijck, O.; Zecchin, A.; Bosseler, L.; Santoni, K.; Planès, R.; Ribo, S.; Fossoul, A.; Gonçalves, A.; Van Gorp, H.; Van Opdenbosch, N.; Van Hauwermeiren, F.; Meunier, E.; Wullaert, A.; Lamkanfi, M. GSDMD drives canonical inflammasome-induced neutrophil pyroptosis and is dispensable for NETosis. *EMBO Rep* **2022**, *23*, e54277.

(24) Sercundes, M. K.; Ortolan, L. S.; Debone, D.; Soeiro-Pereira, P. V.; Gomes, E.; Aitken, E. H.; Condino-Neto, A.; Russo, M.; MR, D. I. L.; Alvarez, J. M.; Portugal, S.; Marinho, C. R.; Epiphany, S. Targeting Neutrophils to Prevent Malaria-Associated Acute Lung Injury/Acute Respiratory Distress Syndrome in Mice. *PLoS Pathog* **2016**, *12*, e1006054.

(25) Zhang, S.; Zhang, Q.; Wang, F.; Guo, X.; Liu, T.; Zhao, Y.; Gu, B.; Chen, H.; Li, Y. Hydroxychloroquine inhibiting neutrophil extracellular trap formation alleviates hepatic ischemia/reperfusion injury by blocking TLR9 in mice. *Clin Immunol* **2020**, *216*, 108461.

REVIEWERS' COMMENTS

Reviewer #1 (Remarks to the Author):

I have gone through the authors comments and modified manuscript titled " PRL2 regulates neutrophil extracellular trap formation contributing to severe malaria and acute lung injury". Authors have performed additional experiments and added additional data based on my reviews and most of their experiments are pointing towards the role PRL2 in neutrophil extracellular trap formation which contributes towards lung injury in severe malaria. Although authors still have not identified parasite protein(s) during severe malaria infection or a signalling cascade that triggers this pathway, but I still appreciate their effort and recommend the publication of the manuscript

Reviewer #2 (Remarks to the Author):

I have reviewed the rebuttal, revised manuscript and the additional data submitted by the authors and I am satisfied with their responses.

Reviewer #3 (Remarks to the Author):

The authors have included many improvements in the manuscript. However the consensus in the malaria community is that Pb ANKA infections cannot be called 'malaria' because of significant differences with the human disease. Since there is no human data to back up these findings, the title of the paper should be changed to 'murine malaria'. Additionally, throughout the paper, the model should be called either 'experimental malaria' or 'murine malaria'

● Response to Reviewers

We thank all 3 reviewers for their remaining comments of this manuscript. All comments are valuable and very helpful.

Below are our point-by-point responses to the reviewers' comments (shown in blue).

Reviewer #1:

I have gone through the authors comments and modified manuscript titled "PRL2 regulates neutrophil extracellular trap formation contributing to severe malaria and acute lung injury". Authors have performed additional experiments and added additional data based on my reviews and most of their experiments are pointing towards the role PRL2 in neutrophil extracellular trap formation which contributes towards lung injury in severe malaria. Although authors still have not identified parasite protein(s) during severe malaria infection or a signaling cascade that triggers this pathway, but I still appreciate their effort and recommend the publication of the manuscript.

Response:

We sincerely appreciate the reviewer's positive evaluation of this manuscript.

Reviewer #2:

I have reviewed the rebuttal, revised manuscript and the additional data submitted by the authors and I am satisfied with their responses.

Response: We thank reviewer for the great comments to improve the quality of our manuscript.

Reviewer #3:

The authors have included many improvements in the manuscript. However, the consensus in the malaria community is that Pb ANKA infections cannot be called "malaria"; because of significant differences with the human disease. Since there is no human data to back up these findings, the title of the paper should be changed to "murine malaria". Additionally, throughout the paper, the model should be called either "experimental malaria" or "murine malaria".

Response: We agree that there are differences between human and murine after Plasmodium infection. However, malaria related acute lung injury is a common manifestation of severe malaria, which appears both in human and in mice. Levels

of neutrophil chemokine IL-8, NE, proteinase-3 and circulating NETs are increased in patients with malaria¹. The title of the paper was changed to “PRL2 regulates neutrophil extracellular trap formation which contributes to severe malaria and acute lung injury” as suggested by the editor.

References:

(1) Aitken, E. H. ; Alemu, A. ; Rogerson, S. J. Neutrophils and Malaria. *Frontiers in immunology* **2018**, *9*, 3005.